# Design principles of a microtubule polymerase

Elisabeth A Geyer[1], Matthew P Miller[2], Chad A Brautigam[3], Sue Biggins[2], Luke M Rice[1]*

[1]Departments of Biophysics and Biochemistry, UT Southwestern Medical Center, Dallas, United States; [2]Howard Hughes Medical Institute, Division of Basic Sciences, Fred Hutchinson Cancer Research Center, Seattle, United States; [3]Departments of Biophysics and Microbiology, UT Southwestern Medical Center, Dallas, United States

**Abstract** Stu2/XMAP215 microtubule polymerases use multiple tubulin-binding TOG domains and a lattice-binding basic region to processively promote faster elongation. How the domain composition and organization of these proteins dictate polymerase activity, end localization, and processivity is unknown. We show that polymerase activity does not require different kinds of TOGs, nor are there strict requirements for how the TOGs are linked. We identify an unexpected antagonism between the tubulin-binding TOGs and the lattice-binding basic region: lattice binding by the basic region is weak when at least two TOGs engage tubulins, strong when TOGs are empty. End-localization of Stu2 requires unpolymerized tubulin, at least two TOGs, and polymerase competence. We propose a 'ratcheting' model for processivity: transfer of tubulin from TOGs to the lattice activates the basic region, retaining the polymerase at the end for subsequent rounds of tubulin binding and incorporation. These results clarify design principles of the polymerase.
DOI: https://doi.org/10.7554/eLife.34574.001

*For correspondence:
Luke.Rice@UTSouthwestern.edu

**Competing interests:** The authors declare that no competing interests exist.

## Introduction

Microtubules are dynamic polymers of αβ-tubulin (hereafter: tubulin) that have critical roles in chromosome segregation and intracellular organization (reviewed in *Desai and Mitchison, 1997*). The polymerization dynamics of microtubules are regulated by multiple cellular factors. Evolutionarily conserved Stu2/XMAP215 proteins (*Gard and Kirschner, 1987*; *Ohkura et al., 1988*; *Wang and Huffaker, 1997*) regulate microtubule dynamics by making microtubules grow faster. Fungal members of this family like Stu2 (from *S. cerevisiae*) or Alp14 and Dis1 (from *S. pombe*) are homodimers, whereas orthologs in higher eukaryotes (e.g. Zyg-9 from *C. elegans*, XMAP215 from *X. laevis*, ch-TOG from *H. sapiens*, and others) are monomeric (*Al-Bassam and Chang, 2011*). Whether monomeric or dimeric, these essential proteins all use multiple tubulin binding tumor overexpressed gene (TOG) domains to localize to the growing microtubule end and promote elongation (*Al-Bassam et al., 2012*; *2006*; *Brouhard et al., 2008*; *Roostalu et al., 2015*; *Widlund et al., 2011*).

How these polymerases make microtubules grow faster remains poorly understood. Studies of XMAP215 (*Brouhard et al., 2008*) revealed that the polymerase (i) is a catalyst that increases the rate of microtubule elongation but not the apparent equilibrium, (ii) acts processively by performing multiple rounds of tubulin incorporation while on the growing end, and (iii) requires a minimum of two TOG domains. In addition to the TOGs, Stu2/XMAP215 family polymerases contain a basic region that is thought to provide lattice-binding affinity (*Wang and Huffaker, 1997*; *Widlund et al., 2011*). Prior studies from our group revealed that the TOG1 and TOG2 domains from Stu2 each bind preferentially and with comparable affinity to curved tubulin (*Ayaz et al., 2012*, *2014*), a conformation that is not compatible with the straight conformation of tubulin in the microtubule lattice.

Based on these and other findings, we speculated that TOG1 and TOG2 function interchangeably, and we proposed a conceptual model in which the catalytic nature of polymerase activity is explained by TOG-mediated 'tethering' that concentrated unpolymerized tubulin near the microtubule end (*Ayaz et al., 2014*). However, a rigorous in vitro test of whether the TOGs are truly interchangeable has not been performed, and it remains unresolved how the basic region contributes to activity and how processivity is achieved.

In the present study, we sought to address these poorly understood aspects of the polymerase mechanism by using in vitro reconstitution assays to determine and quantify how polymerase activity was affected by varying the number/identity of TOGs, by altering the charge of the basic region, and by perturbing the tubulin conformational cycle. Polymerase activity does not require different kinds of TOG domain: Stu2 variants with either TOG1 or TOG2 'inactivated' (mutated to essentially abolish interactions with tubulin, resulting in a molecule with only two functional TOG domains) were functional polymerases with maximal activity about half that of wild-type Stu2. We identified an unexpected, antagonistic relationship between the TOGs and the basic region: Stu2 coats the microtubule lattice without detectable end preference when its TOGs are 'empty', but this lattice binding is attenuated and plus-end specificity is restored when at least two TOGs engage unpolymerized tubulins. Tubulin-induced antagonism of lattice binding suggests a ratcheting model that may explain processivity: delivery/release of polymerase-bound tubulins to the microtubule end 'activates' the basic region, which helps maintain the now empty polymerase at the growing end, poised to capture more unpolymerized tubulin for another round of incorporation. Both polymerase activity and tubulin-induced attenuation of basic:lattice interactions were diminished by perturbations that make tubulin:tubulin interactions stronger. These perturbations apparently lead to futile cycling of the polymerase by bypassing the normal requirement for a microtubule end to stimulate release of TOG-bound tubulins. Our findings provide new insights into how the design of the polymerase determines its activity, and reveal that the tubulin conformational cycle plays a more central role in modulating the activity of this family of polymerases than previously appreciated.

## Results

### Stu2 polymerase activity does not require different TOG domains or dimerization

Stu2 forms a homodimer, with each monomer containing (from N to C) a TOG1 domain, a TOG2 domain, a basic region, and a coiled-coil that mediates dimerization (*Figure 1A*). In an earlier study, we found that Stu2 variants with either TOG1 or TOG2 inactivated for tubulin-binding rescued the loss of wild-type Stu2 in a genetic assay, but monomeric forms of Stu2 showed much poorer rescue activity (*Ayaz et al., 2014*). These results suggested that the presence of two functional TOG domains was not always sufficient for full polymerase activity, and consequently that dimerization and/or how the TOGs were linked was essential for function. To investigate more directly how altering the design of Stu2 (number and type of TOGs, and oligomerization state) affects polymerase activity, we developed an all yeast protein in vitro reconstitution assay analogous to the one reported by (*Podolski et al., 2014*). Our assay uses sea urchin axonemes to seed yeast microtubules; the polymerization dynamics of yeast tubulin with different amounts of Stu2-eGFP variants are monitored by total internal reflection fluorescence (TIRF) microscopy (*Figure 1B*, left). Using a slightly shorter construct, and as observed by others (*Podolski et al., 2014*; *Brouhard et al., 2008*), we confirmed that the eGFP tag did not affect polymerase activity (*Figure 1—figure supplement 1A,D*). These assays revealed, consistent with prior work (*Al-Bassam et al., 2006*; *Podolski et al., 2014*), that Stu2 variants track the growing ends of microtubules (*Figure 1B*, right) and show dose-dependent stimulation of polymerization rates, reaching about 6-fold at saturation (*Figure 1C*; *Figure 1—source data 1*).

To determine whether polymerase activity depends on the number or identity of TOG domains (TOG1 versus TOG2), we measured the polymerase activity of Stu2 variants containing R->A mutations (R200A in TOG1, denoted TOG1*; R519A in TOG2, denoted TOG2*) that inactivate the individual TOG domains for tubulin binding (*Ayaz et al., 2014*, *2012*). Stu2(TOG1*-TOG2) yielded ~40% maximal fold-stimulation of elongation compared to wild-type (*Figure 1C,F*); Stu2 (TOG1-TOG2*) yielded ~60% (*Figure 1C,F*). Both variants retained high-affinity end-binding

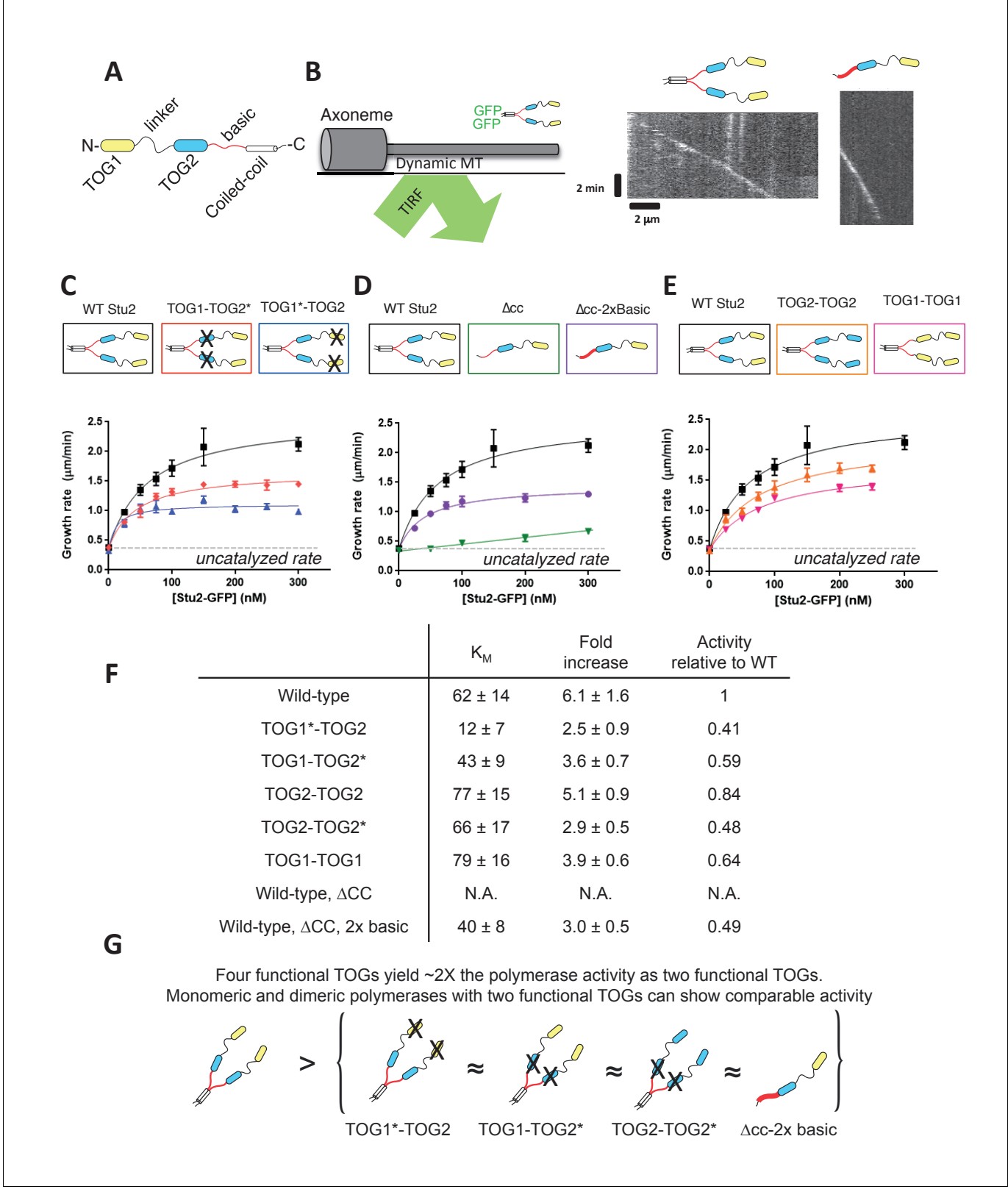

**Figure 1.** Polymerase activity requires at least two TOGs that need not be different, and a basic region; dimerization is not essential. (**A**) Domain organization of Stu2, from N- to C-terminus. (**B**) All-yeast in vitro reconstitution assay. (left) Schematic of the TIRF assay. Axonemes seed unlabeled yeast microtubules in the presence of Stu2-eGFP. (right) Representative kymographs for wild-type Stu2 and for a monomeric Stu2 variant. Microtubule growth rates are determined by tracking the Stu2-eGFP spot on the microtubule end. (**C**) Inactivating TOG1 (blue box/points, TOG1* indicates the R200A

*Figure 1 continued on next page*

*Figure 1 continued*

mutation) or TOG2 (red box/points, TOG2* indicates the R519A mutation) weakens the polymerase activity of the dimer compared to the wild-type dimer (black). Smooth curves indicate a hyperbolic fit to the data. Black: each point represents n = 45 microtubules measured from three different chambers; red/blue: n = 20 microtubules. Error bars are SEM. (D) Eliminating dimerization reduces polymerase activity by sharply weakening end-binding affinity (green). End-binding affinity is restored by increasing the charge in the basic domain (purple). Data were fit as in C., and the WT data are repeated from that panel. Green: n = 45 microtubules measured from three different chambers; purple: n = 25 microtubules. Error bars are SEM. (E) Functional Stu2 variants containing only TOG2 (TOG2-TOG2, orange) or only TOG1 (TOG1-TOG1, pink). Smooth curves indicate a hyperbolic fit to the data. Data were fit as in C., and the WT data are repeated from that panel. Orange: n = 30 microtubules measured from two different chambers; pink: n = 20 microtubules measured. Error bars are SEM. See also *Figure 1—figure supplement 1*. (F) Summary statistics from the hyperbolic fits shown in C, D and E. See also *Figure 1—figure supplement 1*. The maximal activity decreases when the number of TOGs is reduced. Apparent end binding affinity ($K_M$) does not depend strongly on the number of type of active TOGs. (G) Cartoon illustrating that polymerases with two tubulin-binding TOGs are about half as active as WT with its four TOGs.

DOI: https://doi.org/10.7554/eLife.34574.002

The following source data and figure supplements are available for figure 1:

**Source data 1.** Numerical data associated with *Figure 1*.

DOI: https://doi.org/10.7554/eLife.34574.004

**Figure supplement 1.** Additional data supporting the activity of various Stu2 constructs.

DOI: https://doi.org/10.7554/eLife.34574.003

**Figure supplement 1—source data 1.** Numerical data associated with *Figure 1—figure supplement 1*.

DOI: https://doi.org/10.7554/eLife.34574.005

(*Figure 1F*), as evidenced by the hyperbolic activity vs concentration behavior. Inactivating TOG1 gives a slightly stronger effect compared to TOG2, perhaps because TOG1 binds more tightly to tubulin (*Ayaz et al., 2014*). Consistent with this idea, substituting TOG1 with a second copy of TOG2 (denoted Stu2(TOG2-TOG2)) yielded a polymerase that is only slightly weaker than wild-type (5.1-fold maximal stimulation for TOG2-TOG2 compared to 6.1-fold for wild-type) (*Figure 1E,F*, see also *Figure 1—figure supplement 1*; *Figure 1—figure supplement 1—source data 1*). Substituting TOG2 with a second copy of TOG1 (denoted Stu2(TOG1-TOG1)) also yielded an active polymerase, although less so than for TOG2-TOG2 (*Figure 1E,F*). The 'transplanted' TOG1 domain contained ~25 residues outside the core TOG1 domain (see Materials and methods); this extra difference may account for the lower activity of TOG1-TOG1 compared to TOG2-TOG2, and will be discussed in the next section. In summary, different kinds of tubulin-binding TOG domains are not required to support activity, and reducing the number of 'active' TOGs from 4 to 2 reduced polymerase activity approximately 2-fold without weakening end-association.

While these dimeric Stu2 variants with TOG1 or TOG2 inactivated displayed modest decreases in activity but normal end-binding affinity, a monomeric Stu2 variant (Stu2-Δcc, truncated before the coiled-coil segment that mediates dimerization) behaved differently. Stu2-Δcc displayed a substantial decrease in activity as well as greatly reduced end-binding affinity compared to dimeric variants (*Figure 1D,F*). Why did dimeric polymerases operating with two active TOGs (Stu2(TOG1*-TOG2) or Stu2(TOG1-TOG2*)) show robust activity whereas the monomeric Stu2-Δcc polymerase, which also has two active TOG domains, did not? We speculated that reduced basic charge might account for the poor activity of the monomeric variant. Indeed, Stu2-Δcc only has a single basic region but dimeric variants have two (one from each monomer). We therefore measured the activity of Stu2-Δcc-2xBasic, wherein the unstructured basic region was mutated to have twice the normal positive charge, equivalent to what would be found in a dimer. 'Supercharging' the basic region restored high-affinity end-binding (*Figure 1D,F*). The supercharged monomeric polymerase also showed maximal fold-stimulation of elongation rates comparable to the dimeric Stu2(TOG1*-TOG2) or Stu2(TOG1-TOG2*) polymerases that also operate with only two active TOGs (*Figure 1D*).

The experiments described in this section demonstrate that the polymerase activity of Stu2: (i) does not strictly require different kinds of TOG, (ii) is approximately proportional to the number of tubulin-binding TOG domains (when positioned as in wild-type), and (iii) is modulated by the 'strength' of the basic region (*Figure 1G*). The observation that monomeric and dimeric 'two TOG' polymerases show comparable activity as long as their respective basic regions have similar charge indicates that how the active TOGs are linked is of little importance for polymerase activity. Indeed, in Stu2(TOG1-TOG2*) the active TOG1s are linked via the coiled-coil by the natural linker, the

'dead' TOG2 domains, and the basic regions (see TOG1-TOG2* cartoon in *Figure 1C*). Such loose requirements on the elements linking the TOGs suggests that dimeric polymerases like Stu2/Alp14 and monomeric polymerases like XMAP215/chTOG share a common mechanism despite having different oligomerization states. These findings are broadly consistent with the tethering model we proposed previously (*Ayaz et al., 2014*).

## The TOG domain adjacent to the Stu2 basic region controls end localization

Based on the similarity of their structural and biochemical interactions with tubulin, we argued previously that TOG1 and TOG2 were likely to be interchangeable in the polymerase mechanism (*Ayaz et al., 2014*). Here, we observed roughly comparable activity for Stu2(TOG1-TOG2*) and Stu2 (TOG1*-TOG2). To better understand how inactivating TOG1 or TOG2 affects the activity of a polymerase, we sought to measure the specific activity – measured activity normalized to the number of polymerases at the microtubule end – of these variants. Such measurements require quantification of the amount of wild-type or variant Stu2 on the microtubule end.

We first measured fluorescence intensity as a function of concentration for the wild-type Stu2-eGFP spot on the microtubule end, using the same TIRF assay (*Figure 2A*). The intensities showed saturation behavior with increasing concentration (*Figure 2A*). A hyperbolic fit to the concentration-dependent fluorescence yielded a concentration at half-maximal intensity of 20 nM. The concentration-dependence of end-binding for Stu2 is comparable to the concentration-dependence we observed for activity (60 nM, see *Figure 1F*). If we assume (supported by photobleaching analysis, see next section) that the intensity measurements at 5 nM Stu2 reflect individual Stu2 dimers on the microtubule end, then the saturating fluorescence corresponds to ~6 Stu2 dimers on the microtubule end. Thus, the microtubule end can support approximately one Stu2 dimer for every two protofilaments.

Unexpectedly, Stu2(TOG1*-TOG2) and Stu2(TOG1-TOG2*) variants accumulated to different extents on the microtubule end. Whereas Stu2(TOG1*-TOG2) was present in amounts comparable to wild-type, Stu2(TOG1-TOG2*) only reached about half that level (*Figure 2B*; *Figure 2—source data 1*). These differences could reflect the *identity* of the active TOG, the *position* of the active TOG in the primary sequence, or both. Stu2(TOG2-TOG2*) (wherein the natural TOG1 domain was replaced with a TOG2) accumulated to comparable levels as TOG1-TOG2* (*Figure 2B*). Thus, installing a TOG2 domain in the N-terminal position where TOG1 normally resides did not compensate for the inactivation of TOG2 in its natural position. This provides clear evidence that a positional effect – for example proximity to the basic region - contributes to specify the degree of end accumulation. The differential accumulations at the microtubule end means that there are more substantial differences in specific activity than were apparent from our measurements of 'bulk' polymerase activity. Indeed, after normalizing by the different saturating amounts of polymerase on the microtubule end, Stu2(TOG1-TOG2*) shows 20% higher specific activity than wild-type Stu2 (*Figure 2C*), despite having half as many active (non-mutated) TOGs. On a per-active-TOG basis, the specific activity of Stu2(TOG1-TOG2*) is over two-fold higher than wild-type (*Figure 2C*).

Why does inactivating TOG2 for tubulin-binding lead to an increase in specific activity? We speculated that a region of TOG2 outside the conserved tubulin-binding interface might be required for plus-end recognition/localization, and that tubulin binding to TOG2 might antagonize this role. If true, then deleting TOG2 should yield different and stronger effects than mutating TOG2. We therefore prepared Stu2 variants in which the basic-proximal TOG2 domain was deleted (e.g. Stu2(TOG1-ΔTOG2)). Control experiments demonstrated that deleting TOG2 did not compromise the ability of TOG1 to bind curved tubulin, or of the basic region to bind the lattice (*Figure 2—figure supplement 1B*; *Figure 2—figure supplement 1—source data 1*). However, deleting the TOG2 domain completely abolished polymerase activity and end association/tracking (*Figure 2D*, *Figure 2—figure supplement 1A*). This dramatic loss of function could not be rescued by installing a TOG2 domain in place of TOG1 (*Figure 2D*), so the lack of activity must reflect the positioning of the active TOG domain relative to the basic region, not the nature of the active TOG. To more directly test the idea that proximity to the basic domain impacts polymerase activity, we purified a Stu2 mutant that contains a flexible spacer (16 aa, GSSGGSSSGSSGGGSG) between the end of TOG2 (residue 560) and the start of the basic domain (residue 561)(*Figure 2—figure supplement 1D*). The spacer-containing polymerase retained the ability to tip-track and stimulate elongation, but it was substantially less

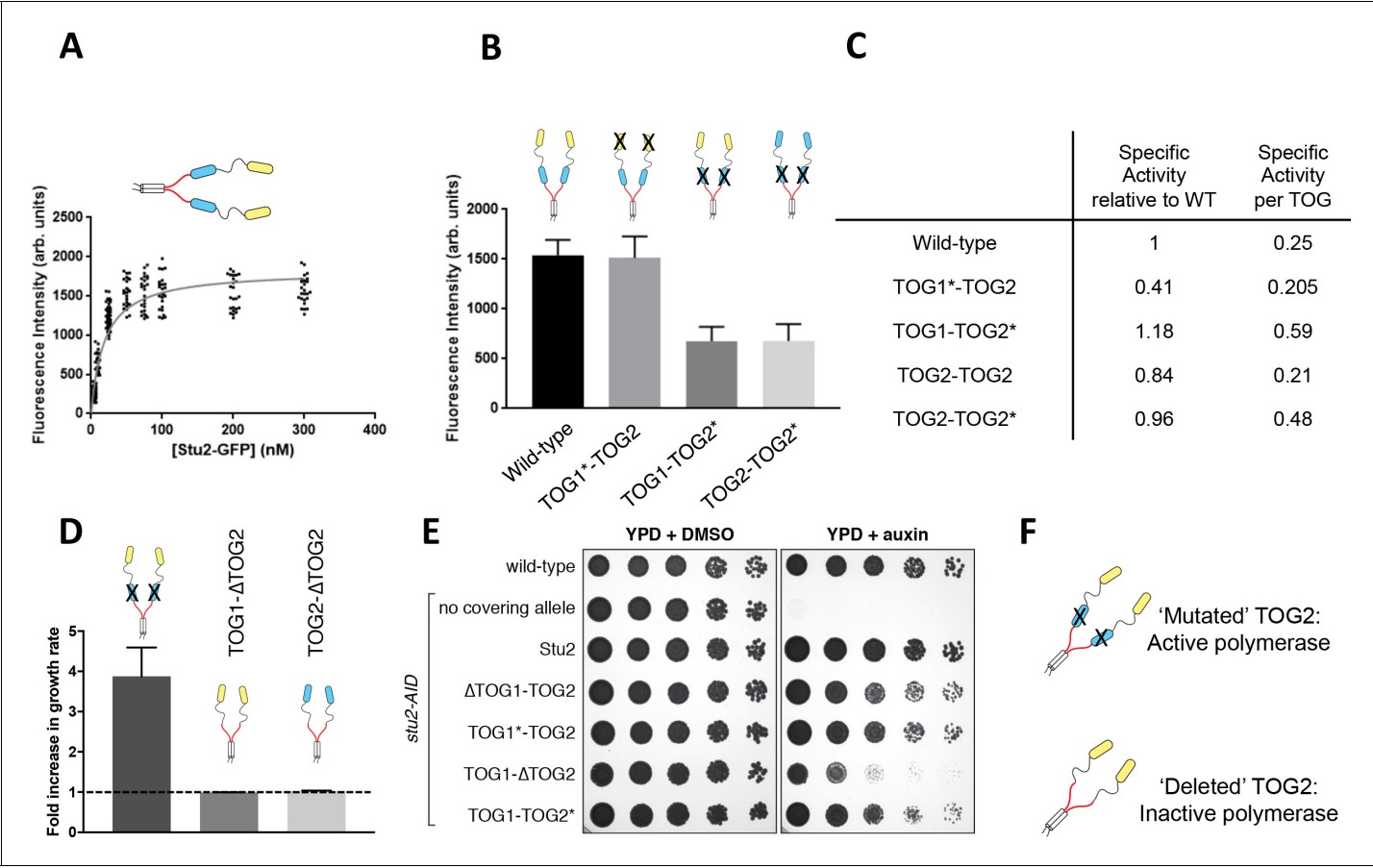

**Figure 2.** A unique functional role for the basic-proximal TOG domain. (**A**) Fluorescence intensity of wild-type Stu2-eGFP dimers on the microtubule end. Intensities were determined as a function of Stu2 concentration by measuring the fluorescent 'spot' at the end of a growing microtubule. The data were fit with a hyperbolic function, yielding a half-maximal concentration of 19 nM, comparable to the half-maximal concentration for activity of 62 nM. N = 200 measured from two different chambers for 5 nM, N = 25 for 10, 50–300 nM, N = 50 measured from two different chambers for 25 nM (**B**) Interfering with tubulin binding by the basic-proximal TOG (TOG2*; R519A mutation) decreases the amount of Stu2 on the microtubule end, but interfering with tubulin binding by the N-terminal TOG (TOG1*; R200A mutation) does not detectably change the amount of Stu2 on the end. The reduced accumulation of TOG1-TOG2* cannot be ascribed to the loss of tubulin binding by TOG2, because TOG2-TOG2* also shows reduced accumulation of the microtubule end. Samples were analyzed with 200 nM Stu2 in the presence of 0.8 μM tubulin in dynamic assays. N = 50 for all. Error bars are SEM. (**C**) Specific activity of Stu2 variants relative to wild-type, on a per-polymerase (left) and per functional TOG (right) basis. Specific activity was obtained by dividing the values from the last column of *Figure 1F* by the relative amount of polymerase on the microtubule end (as determined in *Figure 2B*; one for WT and Stu2(TOG2-TOG2), 0.5 for Stu2(TOG1-TOG2*) and Stu2(TOG2-TOG2*)). Stu2 variants with the basic-proximal TOG inactivated show higher specificity. (**D**) Deleting the basic-proximal TOG domain (ΔTOG2) abolishes polymerase activity and interaction with the microtubule lattice (see *Figure 2—figure supplement 1*), whether the N-terminal TOG domain is TOG1 or TOG2. Changes in polymerase activity may be attributable to the distance between the most basic-proximal TOG and the basic domain (see *Figure 2—figure supplement 1D*) The fold-increase in elongation rate is plotted for different polymerase variants. The dashed line indicates the rate of elongation for the 'no Stu2' control. N = 25 measurements for all; error bars are SEM. Samples contained 0.8 μM tubulin and 200 nM Stu2 variants. (**E**) Wild-type (SBY3), *stu2-AID* (SBY13772) and *stu2-AID* cells expressing various *STU2-3V5* alleles from an ectopic locus (wildtype, SBY13901; TOG1Δ, SBY13904; R200A, SBY13919; TOG2Δ, SBY13907; R519A, SBY13925) were serially diluted (5-fold) and spotted on plates containing either DMSO or 500 μM auxin. Stu2 constructs deleted for TOG1 (ΔTOG1-TOG2) or with TOG1 inactivated for tubulin binding (TOG1*-TOG2) display a mild defect in rescuing the loss of WT Stu2. A Stu2 construct deleted for TOG2 (TOG1-ΔTOG2) shows a more severe loss of rescue activity. A Stu2 construct with TOG2 inactivated for tubulin binding (TOG1-TOG2*) yields full rescue activity. See also *Figure 2—figure supplement 1*. (**F**) Schematic cartoon summarizing that polymerase activity and Stu2 function requires a TOG domain adjacent to the basic region, even if that TOG is compromised for binding to unpolymerized tubulin.

DOI: https://doi.org/10.7554/eLife.34574.006

The following source data and figure supplements are available for figure 2:

**Source data 1.** Numerical data associated with *Figure 2*.

DOI: https://doi.org/10.7554/eLife.34574.008

**Figure supplement 1.** Effects of inactivating, deleting, or replacing the basic-proximal TOG domain.

DOI: https://doi.org/10.7554/eLife.34574.007

*Figure 2 continued on next page*

*Figure 2 continued*

**Figure supplement 1—source data 1.** Numerical data associated with *Figure 2—figure supplement 1*.
DOI: https://doi.org/10.7554/eLife.34574.009

active than wild-type (*Figure 2—figure supplement 1D,E*). These data, together with the data from Stu2(TOG1-TOG1) that also introduced extra sequence between the basic region and the preceding TOG (*Figure 1E*), indicate that proximity of a TOG domain to the basic region is important.

A genetic rescue assay provides additional support for the special nature of the basic-proximal TOG2 domain: deleting TOG2 compromised rescue most severely whereas mutating TOG2, or deleting or mutating TOG1 domain, affected rescue less severely (and comparably to each other) (*Figure 2E*). Stu2 function does not appear to have an absolute requirement for a TOG2 domain next to the basic region, because Stu2(TOG1-TOG1), a construct entirely lacking TOG2 domains, showed appreciable rescue activity (*Figure 2—figure supplement 1C*; slightly different domain boundaries for the transplanted TOG1 were used in the rescue constructs compared to the in vitro experiments described above, see Materials and methods). As in the in vitro assays (*Figure 1E*), in the rescue assay Stu2(TOG1-TOG1) was less active than Stu2(TOG2-TOG2). We conclude that there is a special requirement for a basic-proximal TOG domain, and that there might be some separation of function between TOG1 and TOG2 at this position.

## Processivity and the amount of end resident polymerase together determine maximal achievable activity

The mechanisms of processivity in Stu2/XMAP215 family polymerases are not well understood. We wondered if differences in processivity might explain why Stu2(TOG2-TOG2*) and Stu2(TOG1*-TOG2) differ more than 2-fold in specific activity, despite both using TOG2 as their only tubulin-binding TOG. The hyperbolic fit to the fluorescence intensity of Stu2 on the microtubule end indicates that when the concentration of Stu2 is 5 nM (1/4 of the apparent dissociation constant for tip binding, see *Figure 2A*), there should be on average 1.2 Stu2 dimers on the microtubule end. This suggested that 5 nM Stu2 would be a reasonable concentration for performing single-molecule measurements of Stu2 on the microtubule end. We confirmed using photobleaching analysis that the Stu2 'spots' on dynamic microtubules had intensity comparable to individual Stu2 dimers that bleached in two steps (*Figure 3A*). At this low concentration we observed short 'tracks' of Stu2 fluorescence, with gaps of variable length in between (*Figure 3B*). We fit a one-phase exponential decay to the histogram of end residence times, which yielded a characteristic residence time of 2.2 s (*Figure 3B*, black, 3D; *Figure 3—source data 1*). At the low concentration of 5 nM, Stu2 is not meaningfully affecting microtubule growth rates. We performed 'spike' experiments (5 nM Stu2-GFP + 195 nM unlabeled Stu2) to monitor the behavior of individual Stu2 polymerases under conditions where Stu2 was substantially increasing growth rates. We observed short fluorescent Stu2 'tracks' in these 'spike' experiments, and the distribution of dwell times yielded a characteristic residence time of 1.98 s (*Figure 3B*, orange, 3D). The characteristic residence times for Stu2 at high and low concentrations are quite similar, and are also the same order of magnitude as that measured for XMAP215 (3.8 s). 200 nM Stu2 increases the microtubule elongation rate ~1.8 µm/min over the control (equivalent to 49 tubulins per second), and at this concentration we estimate that there are ~6 Stu2 polymerases on the end. If 6 Stu2s on average account for ~98 tubulins in 2 s (the measured residence time), then each Stu2 adds ~16 tubulins. Thus, like XMAP215, Stu2 is a modestly processive polymerase.

To determine if inactivating TOG1 or TOG2 for tubulin-binding affected end residence, we repeated the same assay using 5 nM of Stu2(TOG1*-TOG2) or Stu2(TOG1-TOG2*). Inactivating the TOG1 domain yielded a modest decrease in dwell times (1.7 s for Stu2(TOG1*-TOG2) compared to 2.2 s for wild-type) (*Figure 3C,D*). In contrast, inactivating the TOG2 domain almost doubled the polymerase dwell time (to 4.0 s for Stu2(TOG1-TOG2*)) (*Figure 3C,D*). This enhanced end residence of Stu2(TOG1-TOG2*) could not be ascribed to some TOG1-specific property, because Stu2(TOG2-TOG2*) also showed a nearly two-fold increase in end-residence time (3.9 s for Stu2(TOG2-TOG2*) compared to 2.2 s for wild-type) (*Figure 3C,D*). Thus, inactivating tubulin binding by the basic-proximal TOG domain actually increases dwell time (and presumably processivity); deleting the TOG2

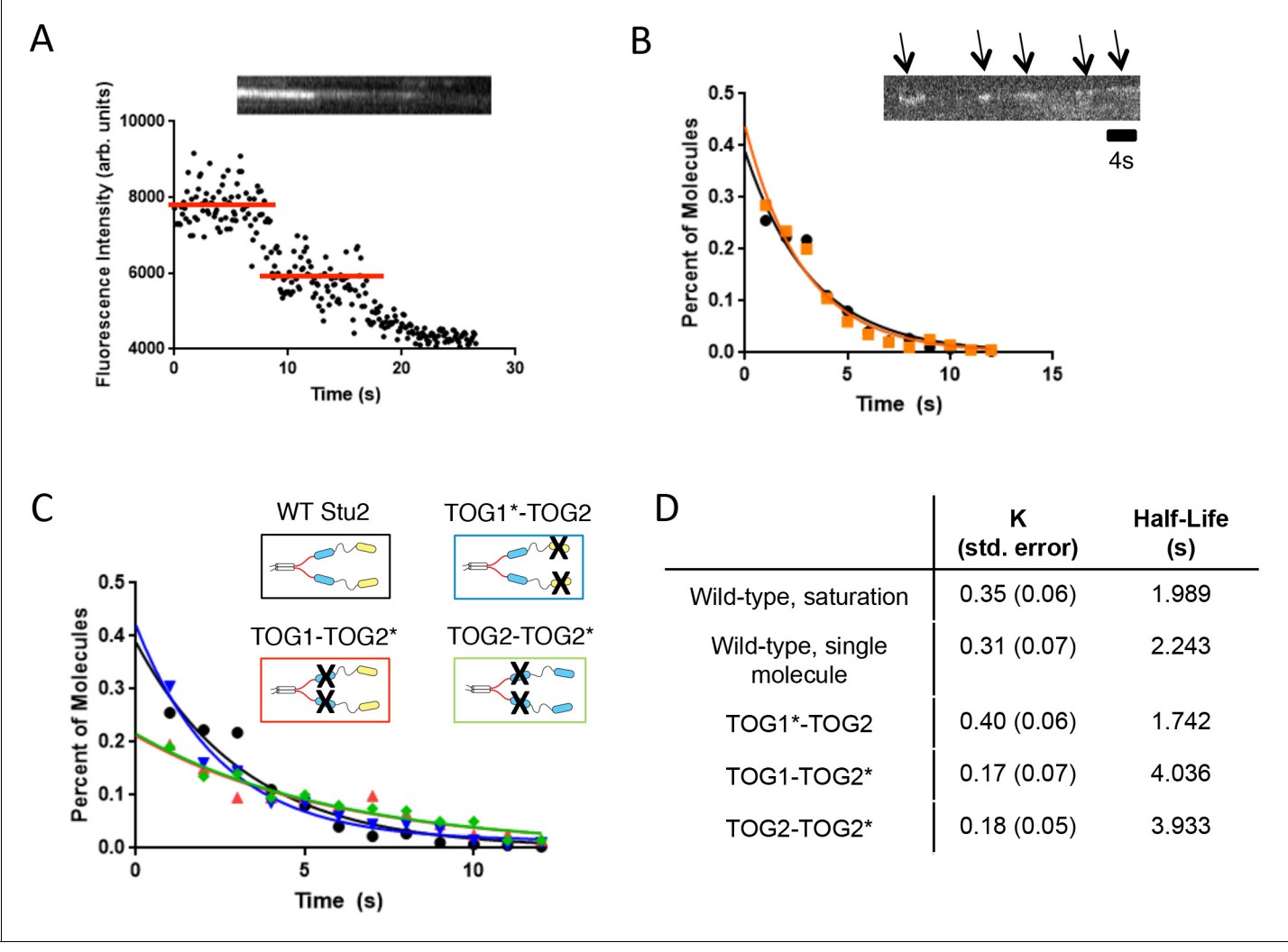

**Figure 3.** Inactivating tubulin binding by the basic-proximal TOG domain enhances processivity of Stu2. (**A**) Example raw fluorescence intensity trace showing two-step photobleaching of a stationary, homodimeric Stu2-eGFP. (**B**) Lifetime distribution for single Stu2-eGFP molecules on the growing microtubule end under single molecule conditions (5 nM Stu2, black) or 'spiked' into a higher concentration reaction (200 nM Stu2 of which only 5 nM is labeled, orange). Inset kymograph illustrates instances of Stu2 'runs' (arrows). Histogram summarizes n = 400 measurements (four independent trials at 1 μM tubulin) for 5 nM Stu2-eGFP and n = 200 measurements (two independent trials at 0.8 μM tubulin) made using 5 nM Stu2-eGFP with 195 nM Stu2-KCK unlabeled. Data, which are plotted as percent of total for comparative purposes, were fit with an exponential, yielding an average residence time of 2.2 s (see also D) for single molecule eGFP and 2.0 s (see also D) for the spike measurements. Scale bar is 4 s. (**C**) Lifetime histograms for Stu2 variants. Compromising tubulin binding by the basic-proximal TOG (TOG1-TOG2*, red trace) yields a roughly 2-fold increase in end residence time. Compromising the N-terminal TOG (TOG1*-TOG2, blue trace) does not increase end-residence time. TOG2-TOG2* (green trace) also shows an increase in end residence time. All samples contained 5 nM Stu2-eGFP variants and 1 μM unlabeled yeast tubulin. Two independent trials of n = 100 measurements for all mutants, yielding a total n = 200. Samples were fit with exponential as done in B to extract average residence times. (**D**) Tabulated summary of results from all exponential fits to residence time distributions in C.

DOI: https://doi.org/10.7554/eLife.34574.010

The following source data is available for figure 3:

**Source data 1.** Numerical data associated with *Figure 3*.

DOI: https://doi.org/10.7554/eLife.34574.011

domain (Stu2(TOG1-ΔTOG2)) abolished end tracking entirely. This counterintuitive behavior is consistent with the idea that TOG2:tubulin engagement antagonizes some other, end-specific function of TOG2; it probably also accounts for the differences in specific activity we measured.

## The strength of interactions between the basic domain and the MT lattice depend on whether TOGs are engaged with αβ-tubulin

Current models for XMAP215/Stu2 family polymerases assume that lattice binding by the basic domain is independent of tubulin binding by the TOGs (e.g [*Ayaz et al., 2014*; *Widlund et al., 2011*]). However, our data demonstrating a need for a TOG domain proximal to the basic region raise the possibility that TOG:tubulin binding may influence basic:lattice interactions, thereby suggesting that TOG:tubulin and basic:lattice interactions may not be independent after all. To begin investigating if basic:lattice interactions are influenced by TOG:tubulin engagement, we prepared a 'doubly dead' Stu2 variant (denoted Stu2(TOG1*-TOG2*)) in which both TOGs were inactivated for tubulin binding. As expected, the doubly dead variant failed to stimulate growth rate (*Figure 4—figure supplement 1*; *Figure 4—figure supplement 1—source data 1*). Unexpectedly, the doubly dead variant robustly coated the lattice, without detectable end preference (*Figure 4*). These results indicate that interactions between TOGs and unpolymerized tubulin attenuate lattice binding by the basic region of Stu2.

Does tubulin-induced attenuation of lattice binding require that all four TOGs engage with tubulin? Dimeric polymerases with TOG1 or TOG2 inactivated (Stu2(TOG1*-TOG2) or Stu2(TOG1-TOG2*)) showed robust tip localization with little binding to the bulk lattice (*Figure 4*). The

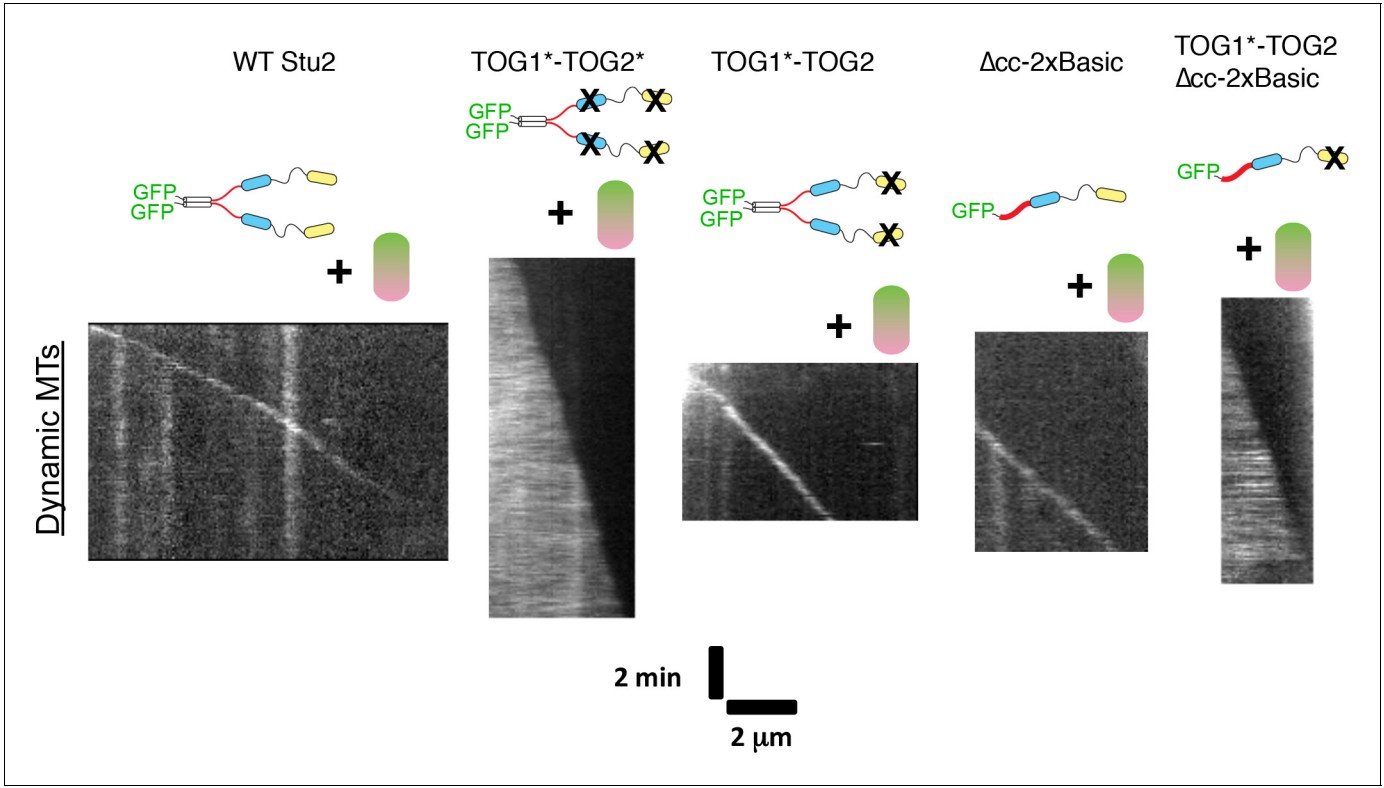

**Figure 4.** Polymerase elements required for preferential plus-end localization of Stu2. Polymerase assays were carried out using wild-type tubulin (0.8 µM) and a panel of polymerase mutants (200 nM each). Representative kymographs are shown for Stu2 variants. Plus-end localization requires at least two tubulin-binding TOGs (WT, TOG1*-TOG2, Δcc-2xBasic) and is not sensitive to how they are linked (compare TOG1*-TOG2 to Δcc-2xBasic). Unexpectedly, polymerases with 0 (TOG1*-TOG2*) or one active TOGs (TOG1*-TOG2-Δcc-2xBasic) robustly coat the body of the microtubule. See also *Figure 4—figure supplement 1*.

DOI: https://doi.org/10.7554/eLife.34574.012

The following source data and figure supplements are available for figure 4:

**Figure supplement 1.** Additional data from assays using mutated TOGs.

DOI: https://doi.org/10.7554/eLife.34574.013

**Figure supplement 1—source data 1.** Numerical data associated with *Figure 4—figure supplement 1*.

DOI: https://doi.org/10.7554/eLife.34574.014

monomeric Stu2-Δcc-2xBasic behaved similarly. However, when one of the two TOGs in this mono-meric construct was mutated (e.g. Stu2(TOG1*-TOG2)Δcc-2xBasic), we observed strong binding to the lattice without detectable tip preference (*Figure 4*); polymerase activity was also lost (*Figure 4—figure supplement 1*). Thus, at least two TOG:tubulin interactions are required to attenuate the lattice binding activity of Stu2.

We next tested a panel of tubulin mutants (*Figure 5A*) to define the tubulin elements required for attenuating lattice-binding by the basic region. We were particularly interested in mutations on tubulin:tubulin interfaces. Such mutants often do not polymerize efficiently (e.g. [*Johnson et al., 2011*] and *Figure 5—figure supplement 1A*), which means we could not apply the 'dynamics' assay we had been using. Consequently, for these assays with tubulin mutants we used GTPγS-stabilized wild-type microtubules as the 'substrate' for lattice binding. Control experiments recapitulated the effects we observed on dynamic microtubules: Stu2 coated the lattice of stabilized microtubules when there was no unpolymerized tubulin present in the assay, and this lattice coating was lost with the addition of unpolymerized wild-type tubulin (*Figure 5B*, left).

We first speculated that the negatively charged C-terminii of TOG-bound tubulins might them-selves bind to the basic region, effectively competing with the microtubule lattice for the basic region. To test this idea, we purified 'tail-less' tubulin lacking the C-terminii of α- and β-tubulin. This tail-less tubulin also inhibited Stu2 binding to the lattice (*Figure 5B*, middle panel). Thus, competing interactions between the basic region and the charged C-terminii of the TOG-bound tubulins cannot explain the tubulin-induced attenuation of lattice binding.

We next considered the possibility that interactions between the TOG-bound tubulins mediate the attenuation of lattice binding. To test if tubulin:tubulin contacts are important, we performed assays with tubulins containing mutations that block or weaken longitudinal (head-to-tail) or lateral (side-to-side) tubulin:tubulin interfaces. Gel filtration (*Figure 5—figure supplement 1B*) and analytical ultracentrifugation binding experiments both show that longitudinal (β:T175R, V179R) and lateral (β:F281A) mutants bind Stu2:TOG1 comparably to wild-type (*Figure 5—figure supplement 1C,D*). Longitudinal interactions between the TOG-bound tubulins do not appear to be important for the tubulin-induced attenuation of lattice binding, because tubulin 'blocked' on its minus-end (T350E α-tubulin; [*Johnson et al., 2011*], *Figure 5B*) or on its plus-end (T175R, V179R β-tubulin; [*Johnson et al., 2011*], *Figure 5—figure supplement 2*) attenuated lattice binding by Stu2 comparably to wild-type tubulin. By contrast, Stu2 remained bound to the lattice in the presence of tubulin carrying mutations at the site of lateral interactions between heterodimers (*Alushin et al., 2014*) (H284A on α-tubulin, *Figure 5—figure supplement 2*; or F281A on β-tubulin, *Figure 5B*, left) Thus, lateral contacts between TOG-bound tubulins are important for the tubulin-induced antagonism of basic:lattice interactions.

The data described in this section reveal that for Stu2, there is antagonistic coupling between TOG:tubulin and basic:lattice interactions: when at least two TOGs on a given polymerase each engage a tubulin, lateral interactions (possibly transient) between those TOG-bound tubulins sub-stantially reduce lattice-binding affinity (*Figure 5C*). We wondered if this antagonism between TOG:tubulin association and lattice binding might represent a more general property of the polymerase family? We obtained mCherry-tagged Zyg-9, the XMAP215/Stu2 family member from *C. elegans* (*Matthews et al., 1998*; *Srayko et al., 2005*), to begin addressing this question. Like Stu2, Zyg-9 coated microtubules in the absence of unpolymerized tubulin, and lattice binding by Zyg-9 was greatly attenuated in the presence of unpolymerized tubulin (*Figure 5D*). Thus, it appears that recip-rocal antagonism between lattice binding and TOGs engaging unpolymerized tubulin is a conserved feature of the Stu2/XMAP215 family of polymerases. This antagonism provides a new way to think about mechanisms of processivity, discussed below.

## Allosteric perturbations that stabilize tubulin:tubulin associations cause futile cycling of the polymerase

Microtubule end recognition and binding to unpolymerized tubulin are mediated by preferential binding of polymerase TOG domains to the curved conformation of tubulin (*Ayaz et al., 2014*, *2012*). However, whether the tubulin conformation cycle impacts polymerase activity in other ways has yet to be investigated. We previously characterized a tubulin 'conformation cycle' mutant, Tub2:T238A (henceforth β:T238A) (*Geyer et al., 2015*). This mutant retains a curved conformation in the unpolymerized state, but it stabilizes microtubules by adopting a more GTP-like conformation in the

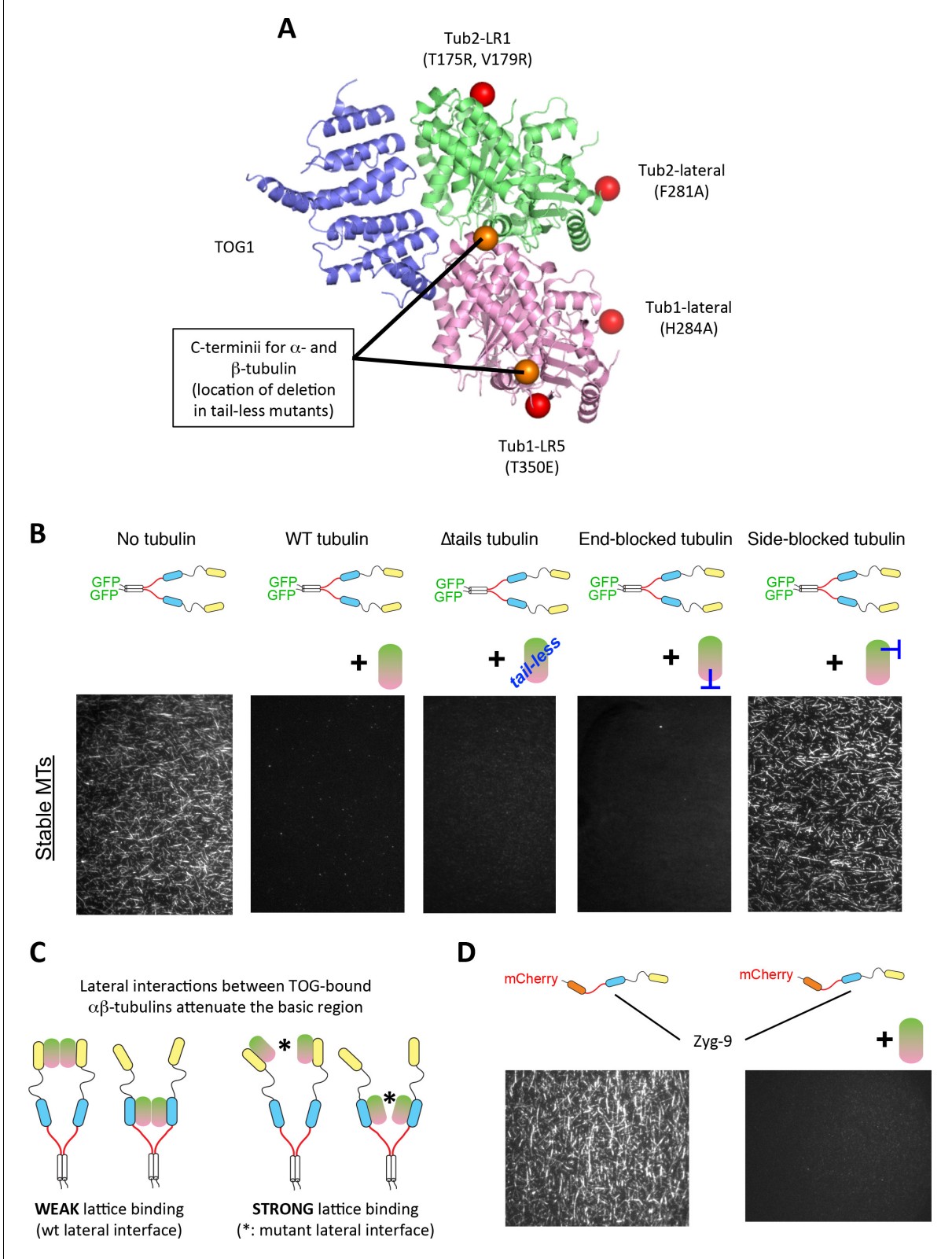

**Figure 5.** Unexpected antagonism between TOG: αβ-tubulin engagement and basic:lattice interactions. (A) The structure of Stu2:TOG1 (slate) bound to yeast αβ-tubulin (α-tubulin in pink and β-tubulin in lime; PDB code 4FFB) is shown in cartoon representation. Red spheres indicate the approximate position of interface blocking mutations on the plus- (β:T175R,V179R) or minus-end (α:T350E) and of perturbing mutations on lateral interaction surfaces (α: H284A, β: F281A). All interface mutations are distant from the TOG-interacting surface. Orange spheres indicate the positions of the structured

*Figure 5 continued on next page*

*Figure 5 continued*

C-terminii of α- and β-tubulin that precede the charged 'tails'. (B) Tubulin elements required to antagonize lattice binding by Stu2. Stu2:microtubule binding assays were monitored by TIRF and performed using stabilized microtubules as the substrate, 100 nM Stu2-eGFP, and 1 µM of tubulin mutants. Stu2 coats the stabilized microtubules when no unpolymerized tubulin in present ('No tubulin'). Lattice-binding is substantially eliminated when wild-type tubulin is included as a competing binding partner ('WT tubulin'). This tubulin-induced antagonism of lattice binding does not require the tubulin tails ('Δtails tubulin') or longitudinal contacts ('end blocked tubulin'). Tubulin perturbed on the lateral interface does not effectively antagonize lattice binding by Stu2 ('side blocked tubulin'), indicating that lateral contacts between TOG-bound tubulins are important. See also *Figure 5—figure supplement 2*. (C) Cartoon illustrating that lateral tubulin interactions (possibly transient) between TOG-bound tubulins antagonize interactions between the basic domain and the MT lattice. Tubulins bound to either the TOG1 or TOG2 domains are illustrated; other combinations of two tubulin-binding TOGs tested yield a similar result. (D) Control of lattice binding by unpolymerized tubulin is not an idiosyncratic property of Stu2. Zyg-9, a monomeric Stu2 family polymerase from *C. elegans*, also shows this tubulin-induced attenuation of microtubule lattice binding. 50 nM Zyg-9-mCherry was used with unlabeled GTPγS-stabilized yeast microtubules; 5 µM bovine tubulin was the competing binding partner.

DOI: https://doi.org/10.7554/eLife.34574.015

The following figure supplements are available for figure 5:

**Figure supplement 1.** Assays characterizing the assembly and TOG-binding properties of tubulin with a mutated lateral interface.

DOI: https://doi.org/10.7554/eLife.34574.016

**Figure supplement 2.** Additional data from assays using mutated tubulins.

DOI: https://doi.org/10.7554/eLife.34574.017

GDP lattice. These effects are reminiscent of how the small molecule taxol allosterically stabilizes microtubules (*Alushin et al., 2014*). Because taxol has been observed to potentiate the activity of XMAP215 (*Zanic et al., 2013*), we examined whether Stu2 would likewise be a more potent polymerase on β:T238A microtubules.

To our surprise, Stu2 barely stimulated elongation and showed weak to no plus-end specificity on dynamic β:T238A microtubules (*Figure 6A,B*; *Figure 6—source data 1*). Instead, Stu2 coated the lattice of these mutant microtubules, reminiscent of what we observed for the 'doubly dead' polymerase Stu2(TOG1*-TOG2*), the TOGs of which are inactivated for tubulin binding. In this case, however, the mutation in β-tubulin does not cause a defect in the interactions between individual TOGs and β:T238A tubulin: isolated TOG1 or TOG2 domains bind to wild-type and to β:T238A tubulin with comparable affinity (*Geyer et al., 2015*). Thus, the β:T238A mutation must antagonize the polymerase activity and preferential end binding of Stu2 through some mechanism that does not entail weakened interactions with the individual TOGs.

We had expected based on prior XMAP215/taxol experiments (*Zanic et al., 2013*) that Stu2 would be a better, not worse, polymerase on β:T238A microtubules. Perhaps the β:T238A mutation in yeast tubulin does not faithfully mimic the effects of taxol binding? We addressed this possibility using Epothilone B, a natural product that affects yeast microtubule structure and stability similarly to the way taxol (which does not bind to yeast microtubules) acts on mammalian microtubules (*Howes et al., 2017*). Polymerase assays using wild-type yeast tubulin with a low concentration (4 µM) of Epothilone B yielded results very similar to those obtained with β:T238A tubulin: Stu2 coated the lattice, and polymerase activity and plus-end tracking were greatly reduced (*Figure 6A,B*). Thus, two independent perturbations that promote tubulin:tubulin interactions increase lattice binding by Stu2 while decreasing its polymerase activity. We infer that normal polymerase function requires not only that the TOG-bound tubulins be curved, but also that they resist self-association-induced straightening enough to ensure that their release from the polymerase occurs only at the microtubule end (*Figure 6C*).

## Discussion

Stu2 variants with TOG1 or TOG2 as the only active TOGs (i.e. competent to bind tubulin) were each functional polymerases. Thus, the quantitative measurements of polymerase activity reported here show that while polymerase activity requires at least two TOG domains, the two TOGs need not be different types. Monomeric Stu2 variants with a 'dimer equivalent' basic domain stimulate elongation to a comparable extent as homodimeric variants with an equal number of tubulin-binding TOG domains, so polymerase activity also does not depend strongly on oligomerization state. That monomeric and dimeric Stu2 variants can have comparable activity indicates that polymerase activity

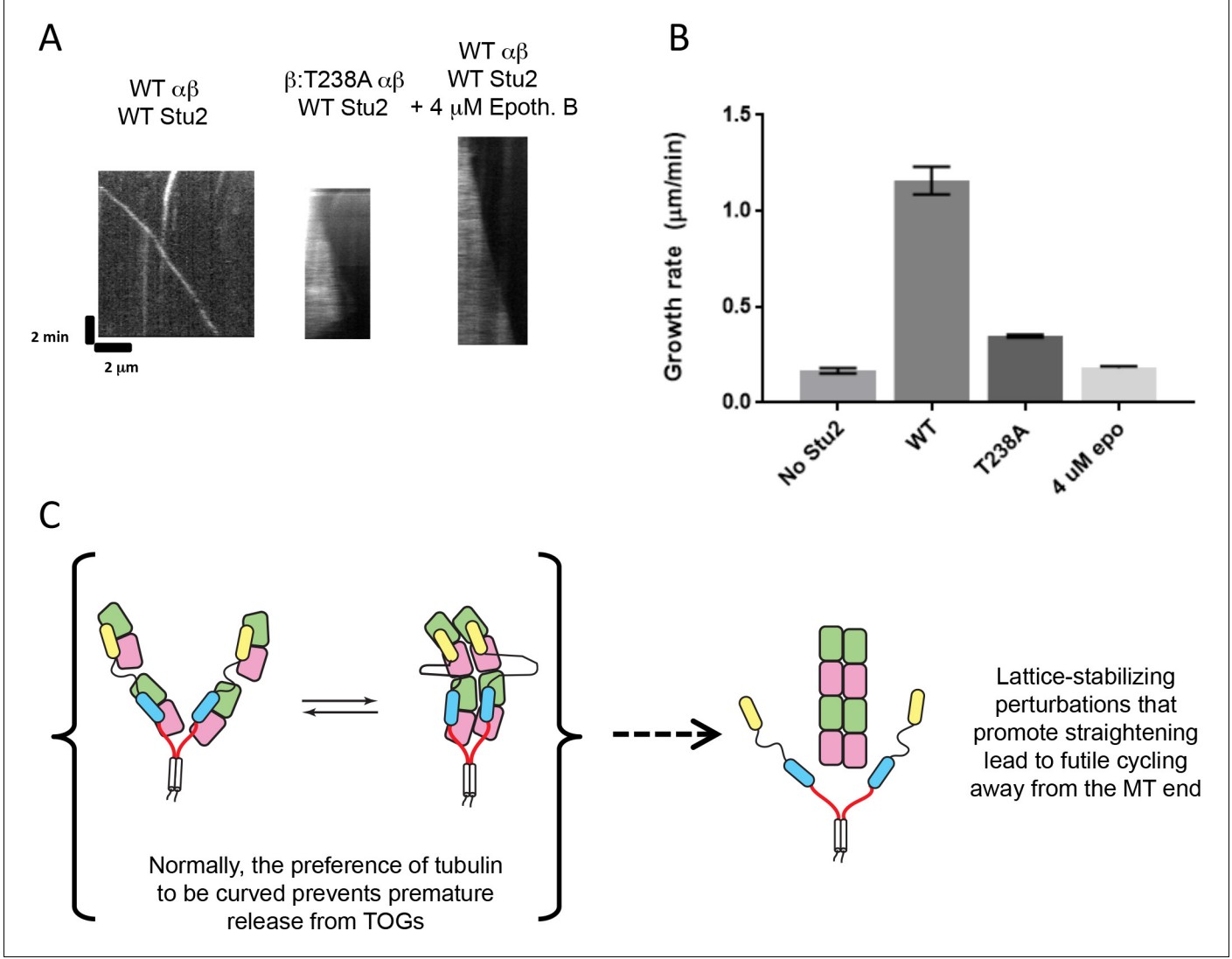

**Figure 6.** Mutation- or drug-induced perturbations that strength tubulin interactions result in loss of Stu2 polymerase activity and tip-tracking. (**A**) Representative kymographs showing normal end tracking, with little lattice binding by Stu2 with wild-type yeast tubulin (left). End-tracking is lost, and lattice binding is enhanced, when β:T238A mutant tubulin was used (right; this mutant stabilizes tubulin:tubulin interactions). Loss of end-tracking and gain in lattice binding also occurred in the presence of wild-type tubulin plus the microtubule-stabilizing drug Epothilone-B (right). Kymographs contain 100 nM Stu2-eGFP and 500 nM yeast tubulin (wild-type or mutant as indicated). 4 μM Epothilone B was added to right panel. (**B**) Stu2 shows reduced polymerase activity in the presence of Epothilone-B or β:T238A mutant tubulin. Measurements were made with 500 nM tubulin (either wild-type or T238A) and 200 nM Stu2-eGFP. N = 25 measurements for all. All error bars are SEM. (**C**) Cartoon illustrating a 'futile cycling' to explain the gain in lattice binding associated with perturbations that strengthen tubulin:tubulin interactions. The preference of tubulin to be curved normally prevents release from TOGs away from the microtubule end. When tubulin:tubulin interactions are strengthened by drug- or mutation-induced perturbation, straightening becomes easier and Stu2 can prematurely release its TOG-bound tubulins away from the microtubule end.
DOI: https://doi.org/10.7554/eLife.34574.018

The following source data is available for figure 6:

**Source data 1.** Numerical data associated with *Figure 6*.
DOI: https://doi.org/10.7554/eLife.34574.019

is largely insensitive to the way that the TOGs are linked. Indeed, in the monomeric Stu2-Δcc-2xBasic variant TOG1 is linked to TOG2 by the natural ~70 amino acid linker. However, in the dimeric Stu2 (TOG1-TOG2*) variant the two active TOG1s are linked by two much longer segments that encompass the natural linker, the mutated TOG2 domain, and the ~100 amino acid basic region that

connects to the coiled-coil. These new findings are broadly consistent with the tethering-based mechanism we proposed previously (*Ayaz et al., 2014*).

We made multiple observations that were not predictable from structural and biochemical properties of isolated TOGs. These include (i) the 'positional' separation of function between the N-terminal and basic-proximal TOG domains in Stu2, (ii) the antagonism between TOG:tubulin and polymerase:lattice binding, and (iii) the loss of polymerase activity that accompanies perturbations that strengthen tubulin:tubulin interactions. These findings provide new insights into the molecular logic and functional design of these polymerases, and their implications for mechanism will be discussed in subsequent sections.

## A positional separation of function is important for polymerase activity

Stu2(TOG1*-TOG2) and Stu2(TOG1-TOG2*) were present on the microtubule end in different amounts (*Figure 2B*), showed different processivity (*Figure 3C,D*), and had different specific activity (*Figure 2C*). Polymerase activity was completely lost when the basic-proximal TOG2 domain was deleted (TOG1-ΔTOG2 or TOG2-ΔTOG2), even though inactivating TOG2 mutations were tolerated (Stu2(TOG1-TOG2*) or Stu2(TOG2-TOG2*)). Our data clearly demonstrate position-dependent requirements for TOG function: the loss of polymerase activity caused by deletion of the TOG2 domain (TOG1-ΔTOG2) was not rescued by replacing TOG1 with TOG2 (TOG2-ΔTOG2). There may also be a weaker separation of function along TOG identity: Stu2(TOG1-TOG1), an 'all TOG1' variant, was less active than Stu2(TOG2-TOG2) in vitro and in genetic rescue assays (*Figure 1E*). That polymerase activity necessitates a TOG domain next to the basic region – even if that TOG has been inactivated for tubulin binding – indicates that some yet-to-be-determined distinctive feature there is important for polymerase activity.

Deletion of the TOG2 domain eliminated plus-end selective localization (*Figure 2D*) in addition to abolishing polymerase activity (*Figure 2—figure supplement 1A*). Yet Stu2(TOG1-ΔTOG2) retains what we thought should have been the minimal requirements for plus-end localization: TOG domains that bind preferentially to the curved conformation of tubulin (two TOG1s in the case of Stu2(TOG1-ΔTOG2); Stu2(TOG2-ΔTOG2) polymerases also did not localize to the plus end) and a basic region that provides lattice-binding affinity. Stu2 variants that only contained a single 'active' TOG domain (e.g. Stu2(TOG1*-TOG2)-Δcc-2xBasic) also failed to show plus-end specific localization or polymerase activity. Thus, plus-end specific localization requires at least two active TOGs and interactions with unpolymerized tubulin, the same requirements for polymerase activity. The simplest explanation for these shared minimal requirements is that plus-end specific localization actually depends on polymerase activity.

## Unpolymerized tubulin controls lattice binding by the basic region

A synthetic polymerase can be constructed from TOG1-TOG2 and an unrelated basic element (*Widlund et al., 2011*). For this and other reasons, the basic region has been assumed to be an independently acting appendage to the TOGs. However, under otherwise identical conditions, we showed that 'empty' polymerases (no tubulin bound to the TOGs) were robustly recruited along the entire length of the microtubule but 'full' polymerases (TOGs engaged with unpolymerized tubulin) only bound at the tip. That lattice binding is antagonized by TOG:tubulin interactions represents an unexpected design principle of the polymerase that appears to be conserved in polymerases from higher eukaryotes (*Figure 5D*). These findings indicate that there is a more intimate relationship between the TOGs and the basic region than previously thought, but we can only speculate about the underlying molecular mechanism. Tubulin-induced antagonism of lattice binding requires interactions between the TOG-bound tubulins (*Figure 5B,C*); it seems possible that these tubulin:tubulin interactions could either alter the overall conformation of Stu2 or provide a hybrid binding site that can compete for the basic region. Other models are possible, and more work will be required to unambiguously define the mechanism.

Stu2 and XMAP215 can both diffuse on the microtubule lattice in a way that depends on their basic regions and on the negatively charged C-terminal tails of α- and β-tubulin (*Brouhard et al., 2008*; *Podolski et al., 2014*), but our data now show that these lattice interactions are attenuated when the TOGs are engaged with unpolymerized tubulins. Thus, compared to empty polymerases, polymerases carrying tubulins must be specifically disadvantaged for diffusing on the lattice. This in

turn means that polymerases diffusing to the microtubule end will for the most part arrive 'empty'. Consequently, diffusive 'tubulin shuttling' to the end will make little contribution to polymerase activity.

## Futile cycling of the polymerase occurs when tubulin self-assembly contacts are too strong

Surprisingly, mutation- or drug-induced perturbations that promote tubulin self-association and 'straightening' (*Geyer et al., 2015*; *Elie-Caille et al., 2007*; *Bode et al., 2002*) led to a substantial reduction in maximal polymerase activity. These perturbations also caused Stu2 to coat the microtubule lattice in the presence of unpolymerized tubulin. This concomitant loss of plus-end localization and polymerase activity provides additional evidence consistent with the idea that end localization and polymerase activity are inseparable. Whether in the presence of epothilone and wild-type tubulin, or in the presence of β:T238A tubulin, Stu2 behaved as if its TOG domains were empty. Yet both TOG1 and TOG2 bind to β:T238A tubulin with comparable affinity as they bind to wild-type (*Geyer et al., 2015*), so some other mechanism must account for the loss of polymerase activity with these perturbations.

The ability of a single polymerase to bind multiple tubulins results in a high local concentration of tubulin. How might perturbations that favor tubulin self-association actually reduce polymerase activity? Under normal circumstances (wild-type tubulin without Epothilone present), even the high local concentration of TOG-bound tubulins is presumably not sufficient to overcome the energetic cost of tubulin straightening. This barrier to straightening must help prevent premature tubulin release from TOGs away from the microtubule end. Consequently, the preference of tubulin to be curved antagonizes polymerase binding along the body of the microtubule. By contrast, in the presence of mutation- or drug-induced perturbations that strengthen tubulin self-association and straightening, Stu2 behaves as if it were empty: it binds the lattice without detectable end preference. Thus, diminishing the energetic cost of self-association-induced tubulin straightening must lead to nonproductive release of polymerase-bound tubulins away from the microtubule end. That this nonproductive release also eliminates preferential end binding by Stu2 reveals an unanticipated link between polymerase mechanism and the tubulin conformation cycle. If tubulin straightens too easily, polymerase activity is lost to futile cycling, in which straightening-induced release from TOGs is no longer restricted to the microtubule end.

## Concluding remarks

Stu2/XMAP215 family polymerases have been compared to formins (*Brouhard et al., 2008*), which are unrelated polymerases that promote fast elongation of actin filaments (reviewed in [*Goode and Eck, 2007*]). This comparison no longer seems apt. For the highly processive formins, end localization can be separated from polymerase activity (*Li and Higgs, 2003*; *Pruyne et al., 2002*; *Kovar et al., 2006*; *Otomo et al., 2005*). However, in modestly processive Stu2, end localization and polymerase activity appear to be inseparable. This and other findings about Stu2 are reminiscent of a different actin polymerase, Ena/VASP. Indeed, Ena/VASP proteins enhance actin elongation through a tethering mechanism (*Breitsprecher et al., 2011*; *Hansen and Mullins, 2010*; *Winkelman et al., 2014*) that uses WH2-like domains, some of which operate analogously to TOGs because they bind preferentially to unpolymerized (actin) subunits (*Bachmann et al., 1999*; *Hüttelmaier et al., 1999*). Ena/VASP proteins share three other functional characteristics with Stu2 that formins do not: (i) Ena/VASP proteins are only weakly processive, (ii) their polymer 'side binding' is attenuated in the presence of unpolymerized monomers, and (iii) their end localization requires interactions with unpolymerized monomers (*Hansen and Mullins, 2010*). For both Stu2 and for Ena/VASP, it seems that processivity emerges from the ability of the polymerase to alternate between stronger and weaker states of lattice association in a way that is controlled by whether or not unpolymerized subunits are bound (*Figure 7*; see also [*Hansen and Mullins, 2010*; *Breitsprecher et al., 2011*]). While for Stu2 it is clear that tethering-based transfer of monomers from TOGs to the polymer end can promote faster elongation, some other post-delivery mechanism is required to maintain the polymerase at the microtubule end for subsequent rounds of delivery. Ratcheting by alternating engagement between TOGs and the basic region is a design principle of the polymerase that provides a simple conceptual model to explain its processive action.

**Ratcheting, alternating engagement model for processivity**

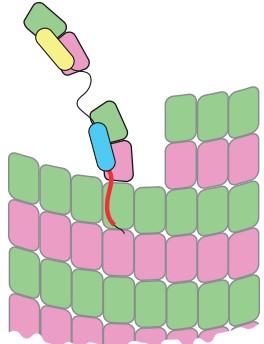

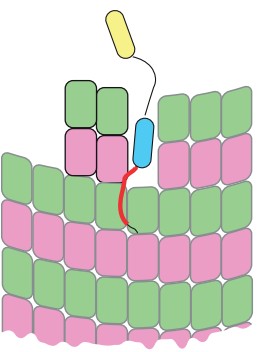

"Pre-release", loaded TOGs:
strong TOG:αβ-tubulin binding
weak basic:lattice binding

"Post-release", empty TOGs:
no TOG:αβ-tubulin binding
strong basic:lattice binding

**Figure 7.** Speculative 'ratcheting' model for processivity. The model was inspired by the fact that the TOG:tubulin engagement status regulates the lattice-binding activity of the basic region. We assume two limiting states for Stu2 on the microtubule end: a 'loaded' state wherein the basic region binds weakly to the lattice because at least two TOGs are engaged with tubulins, and an 'empty' state wherein the basic region binds strongly to the lattice because the TOGs have released their tubulins to the microtubule. In this 'post-release' state, strong interactions between the basic region and the lattice help retain the empty polymerase near the growing end where it is poised to capture 'fresh' unpolymerized tubulins for another round of activity. Capture and incorporation of fresh tubulins drives movement with the growing end.
DOI: https://doi.org/10.7554/eLife.34574.020

## Materials and methods

### Protein expression and purification

Plasmids to express wild-type and β:T238A yeast αβ-tubulin were previously described (*Johnson et al., 2011*; *Ayaz et al., 2012*; *Geyer et al., 2015*). Plasmids to express the β:F281A mutation of Tub2p (yeast β-tubulin) and α:H284A mutation of Tub1p (yeast α-tubulin) were made by QuikChange (Stratagene) mutagenesis, using expression plasmids for wild-type Tub2 and Tub1 as template and with primers designed according to the manufacturer's instructions. Three Stu2p constructs, in pHAT vector containing N-terminal H6 tag, C-terminal eGFP-tag followed by a *Strep*-tag II, were a gift from Dr. Gary Brouhard: Stu2 residues 1–658 (Δcc; monomer); Stu2 residues 1–761 (Δtail; dimer, no C-terminal tail); Stu2 residues 1–888 (WT, full-length; dimer). Mutations were introduced into the monomeric (658) or fully dimeric (888) background using QuikChange (Stratagene) mutagenesis or traditional restriction-based cloning after PCR. The TOG1-TOG1 Stu2FL plasmid was made using NEBuilder HiFi DNA Assembly (New England Biolabs, Ipswich, MA). A gene block (gblock) encompassing TOG1 (residues 1–272) and the Stu2 basic domain (560-661) was purchased from IDT DNA (Coralville, IA). The 'transplanted' region encompasses the structured TOG1 domain with a C-terminal extension of ~20 additional amino acids from the linker sequence; shorter constructs that did not include this linker sequence proved to be unstable upon purification. The parent plasmid, excluding the region coding for the to-be-replaced TOG2 sequence, was amplified in a second reaction with primers sharing overlap to the gene block. The two products were mixed,

incubated and transformed according to the manufacturer's instructions. The integrity of all expression constructs was confirmed by DNA sequencing.

Wild-type and all tubulin mutants (β:T238A, β:T175R/V179R β:F281A, α:H284A, α:T350E) yeast αβ-tubulin were purified from inducibly overexpressing strains of *S. cerevisiae* using Ni-affinity and ion exchange chromatography, as previously described (*Johnson et al., 2011*; *Ayaz et al., 2012*, *2014*; *Geyer et al., 2015*). Tubulin samples were stored in storage buffer for dynamics assays (10 mM PIPES pH 6.9, 1 mM $MgCl_2$, 1 mM EGTA) containing 50 μM GTP or storage buffer for analytical ultracentrifugation experiments (25 mM Tris pH 7.5, 100 mM NaCl, 1 mM $MgCl_2$, 1 mM EGTA) containing 20 μM GTP. Expression of all Stu2p constructs, wild-type and mutant, were induced in *E. coli* using Arctic Express Cells with N-terminal $His_6$ tags and C-terminal eGFP and Strep-tag II. Samples were induced with 0.5 mM IPTG for 24 hr at 10°C. Cell pellets were resuspended in lysis buffer (50 mM $Na_2HPO_4$, 300 mM NaCl, 40 mM imidazole, 5% glycerol) and sonicated for 30 min in the presence of PMSF before clarification by centrifugation. Cleared lysate was loaded onto a His60 Superflow Column (Clontech) and the final sample was eluted in lysis buffer containing 300 mM imidazole. Pooled elution fractions containing Stu2 were loaded onto a 3 mL Strep-Tactin Superflow column (IBA, Germany) and eluted in RB100 (25 mM Tris pH 7.5, 100 mM NaCl, 1 mM $MgCl_2$, 1 mM EGTA) containing 5 mM desthiobiotin. For storage, final samples were exchanged into RB100 with 2 mL, 7K MWCO Zeba spin desalting columns (Thermo Scientific). Expression of purification of TOG1 domain from Stu2 ('TOG1-tail', 1–317) was previously described (*Ayaz et al., 2012*, *2014*).

## In Vitro reconstitution assays using TIRFM

Flow chambers were prepared as described previously (*Gell et al., 2010*), with the exception that sea urchin axonemes (*Waterman-Storer, 2001*) were used to seed growth of yeast microtubules. Chambers were rinsed with BRB80 (80 mM PIPES pH 6.9, 1 mM $MgCl_2$, 1 mM EGTA), followed by 10 min incubation with sea urchin axonemes. Chambers were then blocked with 1% F-127 Pluronic in BRB80 for 5 min, and washed with 1X PEM (100 mM PIPES pH 6.9, 1 mM EGTA, 1 mM $MgSO_4$) containing 1 mM GTP. Reaction chambers were sealed with VALAP after addition of Stu2p and αβ-tubulin samples.

Samples of Stu2p wild-type or mutant proteins, along with wild-type or β:T238A yeast αβ-tubulin were prepared in imaging buffer (1X PEM +50 μM GTP +0.1 mg/mL BSA +antifade reagents (glucose, glucose oxidase, catalase) (*Gell et al., 2010*) A subset of samples were performed in the presence of either 50 μM GTPγS in place of GTP, or with the addition of 1 or 4 μM epothilone-B. For the majority of experiments, Stu2 concentration ranged from 25 to 400 nM, while tubulin concentrations were kept constant throughout the experiment (either 500 nM, 800 nM, 1 μM).

Microtubule (MT) dynamics and Stu2 location/signal were imaged by total internal reflection fluorescence (TIRF) microscopy using an Olympus IX81 microscope with a TIRF ApoN 60x/1.49 objective lens, a 491 nm 50 mW solid-state laser and Hamamatsu ORCA-Flash2.8 CMOS camera (Olympus). In assays where Stu2 signal was not detectable on growing MTs or no Stu2 was present in the reaction, MT dynamics were imaged by differential interference contrast microscopy (DIC) using an Olympus IX81 microscope with a TIRF ApoN 60x/1.49 objective lens and DIC prisms. Illumination at 550 nm was obtained by inserting a bandpass filter of 550/100 nm (Olympus) in the light path. Temperature for all assays were maintained at 30°C using a WeatherStation temperature controller with enclosure fit to the microscope's body. Micro-Manager 1.4.16 (*Edelstein et al., 2010*) was used to control the microscope.

In TIRF assays, MT dynamics were recorded by taking an image every 3–5 s for 15–30 min. MT growth rates were measured manually by creating kymographs using the ReSlice plugin for ImageJ (*Schneider et al., 2012*). From the kymographs, MT growth rates were manually measured by taking the length of the MT from the start of the Stu2 tip-tracking region at the MT end to the base of the axoneme, and repeating this measurement at a later time, t. Changes in MT length as a function of time were then calculated. In DIC assays, MT dynamics were recorded by taking an image every 500 ms for 30 min. At the end of each movie, a set of 100 out-of-focus background images was taken for background subtraction. To improve signal to noise, batches of 10 raw images were averaged using ImageJ (*Schneider et al., 2012*) and intensity normalized before background subtraction. MT length was measured manually using a PointPicker plugin for ImageJ. Rates of MT elongation were determined as described previously (*Walker et al., 1988*). Average growth rates as a function of Stu2 concentration, for each given experiment, were then analyzed in GraphPad Prism 7.01 fitting

experimental data with an altered Michaelis-Menten equation set to specify the change in overall fold offsets in growth rate. Fold-offset changes and Km values were fit parameters.

## End intensity dynamic assays

Flow chambers were prepared as described above. Samples of Stu2p (wild-type or mutant) with wild-type yeast αβ-tubulin were prepared in imaging buffer. For all single molecule end-intensity experiments, Stu2 concentration was held at 5 nM and wild-type yeast αβ-tubulin was used at 1–1.5 μM. For measuring end intensity as a function of Stu2 concentration, concentrations of Stu2 between 10 and 300 nM were used; wild-type yeast tubulin concentration was held constant at 800 nM.

MT dynamics and Stu2 location were imaged at 30°C by TIRF microscopy as described above but using an Andor EMCCD iXon (Andor) camera. Images were recorded every 3 s over a period of 10–15 min.

End intensity measurements to determine the number of Stu2 molecules at the end of a MT were all manually measured in ImageJ (*Schneider et al., 2012*). In single molecule (5 nM Stu2eGFP signal tracking) Stu2 assays, all experimental data sets were background subtracted using the Mosaic Background Subtractor plugin for ImageJ (*Schneider et al., 2012*). End intensity values for Stu2 at a growing MT end were measured using the Measure feature in ImageJ (*Schneider et al., 2012*), where the end of the MT was measured and marked by a single pixel. For experiments where populations of Stu2 molecules reside at the growing MT end (Stu2 concentrations above 5 nM), the intensity of the entire Stu2 comet at the end of a growing MT was measured using the Oval marker and Measure feature in ImageJ (*Schneider et al., 2012*).

## Single molecule processivity assays

Flow chambers were prepared and imaged as described above. For 'strict' single molecule processivity experiments, Stu2 concentration was held at 5 nM and wild-type yeast αβ-tubulin was used at 1–1.5 μM. For 'spike' single molecule processivity experiments at higher overall Stu2 concentration, 5 nM Stu2-eGFP was added along with 195 nM unlabeled Stu2-KCK, with 800 nM wild-type yeast αβ-tubulin.

MT dynamics and Stu2 location and processivity were imaged by TIRF microscopy at 30°C using an Andor EMCCD iXon (Andor) as described above. Images were recorded using streaming acquisition with a 100 ms exposure for 30–60 s.

End processivity times of Stu2 proteins were measured manually, by creating kymographs using the ReSlice plugin for ImageJ (*Schneider et al., 2012*). The length of time Stu2 was present at the end of a MT was recorded. In total, 'strict' single molecule wild-type Stu2 dimer experiments were repeated over four independent experiments, with each experiment yielding 100 Stu2 end-time measurements (n = 400). For 'saturation' single molecule assays, experiments were repeated over two independent sets, with each experiment yielding 100 Stu2 end-time measurements (n = 200). For Stu2 mutants, experiments were repeated over two independent sets, with each experiment yielding 100 Stu2 end-time measurements (n = 200). Time measurements were then sorted by the whole second, data were converted to percentage of molecules by diving the number of molecules at each time point by the total number of molecules analyzed in the set. Percentage of molecules data was imported and analyzed using GraphPad Prism 7.01. Data sets were fit with a one phase exponential decay, constraining the fit such that the plateau must be greater than 0.

## Stable MT Flow-In experiments

To prepare stable wild-type yeast MT, wild-type yeast αβ-tubulin was polymerized in the presence of 500 μM GTPγS in assembly buffer (see above). The mixture was incubated for 90 min at 30°C.

Flow chambers were prepared as described above. Stable microtubules were attached to the cover slip using His-Tag Antibody (1:200, Gentech), which was incubated in the chamber for 10 min before blocking with 1% Pluronic F-127 in BRB80 for 5 min, followed by a wash with BRB80. Preformed, GTPγS-stabilized wild-type yeast MTs were incubated in the chamber for 10 min, followed by a wash with BRB80 to remove unbound MTs. Solutions containing a range of Stu2 concentrations, with or without free wild-type yeast αβ-tubulin in imaging buffer were mixed, incubated on ice for 10 min, then introduced into the chamber immediately prior to data collection. MT fields were imaged at 30°C by TIRF microscopy as described above using an Olympus IX81 microscope with a

Hamamatsu ORCA-Flash2.8 CMOS camera (Olympus), to view the presence and location of Stu2-eGFP molecules; fields were also imaged by DIC microscopy as described above. Images of MTs under TIRF and DIC fields were taken every 30 s for about 5 min. MT images were viewed using ImageJ (*Schneider et al., 2012*).

Depolymerization experiments using this assay setup were performed as previously described (*Geyer et al., 2015*), using the Hamamatsu ORCA-Flash2.8 CMOS camera (Olympus). MT depolymerization was measured manually using ImageJ (*Schneider et al., 2012*), using the average rate of MT depolymerization over the course of imaging to determine the depolymerization rate over an hour time frame.

## Binding experiments

Gel filtration binding experiments were performed by loading 500 µL samples of TOG, tubulin wild-type or mutants (α:H284A or β:F281A) or mixtures of TOG:tubulin samples onto a Shodex KW-803 column equilibrated in BRB80 (80 mM PIPES pH 6.9, 1 mM MgCl2, 1 mM EGTA) with 50 µM GTP. Samples tested contained either 1 µM TOG1-tail (1-317), 1 µM tubulin wild-type or mutant alone (α: H284A or β:F281A), and 1 µM TOG1-tail with 1 µM tubulin wild-type or mutant (α:H284A or β: F281A). All samples were prepared, allowed to equilibrate on ice for 20 min and then loaded onto the column.

Samples for analytical ultracentrifugation (Stu2 TOG1-tail, wild-type yeast αβ-tubulin, 'long-blocked' yeast αβ-tubulin mutant β:T175R/V179R, 'side-blocked' yeast αβ-tubulin mutant, β: F281A) were dialyzed into final buffer conditions of RB100 (25 mM Tris pH 7.5, 1 mM MgCl$_2$, 1 mM EGTA, 100 mM NaCl) containing 20 µM GTP with a 2 mL, 7K MWCO Zeba spin desalting columns (Thermo Scientific). The samples shown in *Figure 5—figure supplement 1* contain 0.3 µM WT yeast αβ-tubulin and 0.1, 0.3, 1.2, 3 µM TOG1-tail or 0.3 µM β:T175R/V179R or β:F281A yeast αβ-tubulin and 0.1, 0.3, 0.9, 1.5, 3 µM TOG1-tail. Samples were mixed and incubated at 4°C for at least one hr prior to the experiment. All analytical ultracentrifugation experiments were carried out in an Optima XL-I centrifuge using an An50-Ti rotor (Beckman-Coulter). Approximately 390 µL of each sample were placed in charcoal-filled, dual-sector Epon centerpieces. Sedimentation (rotor speed: 50,000 rpm) was monitored using absorbance at 229 nm and centrifugation was conducted at 20°C after the centrifugation rotor and cells had equilibrated at that temperature for at least 2.5 hr. Protein partial-specific volumes, buffer viscosities, and buffer densities were calculated using SEDNTERP (*Laue et al., 1992*). Data were analyzed using the *c(s)* methodology in SEDFIT (*Schuck, 2000*; *Schuck et al., 2002*). The distributions were integrated in GUSSI (*Brautigam, 2015*) and analyzed in SEDPHAT (*Schuck, 2010*) with a 1:1 effective-particle model, fixing the *s*-values of the αβ-tubulins and the TOG1-tail and allowing the *s*-value of the αβ-tubulin:TOG1 complex to refine.

## Microtubule pelleting assays

To test the ability of lateral yeast tubulin mutants (α:H284A or β:F281A) to form MTs, experiments were performed as previously described (*Geyer et al., 2015*). Briefly, samples containing 1 µM of a single yeast tubulin variant, either wild-type Tub2-H6, wild-type internal Tub1-H6, Tub1:H284A or Tub2:F281A, were incubated in assembly buffer (100 mM PIPES pH 6.9, 10% glycerol, 2 mM MgSO4, 0.5 mM EGTA, 50 uM GTP) at 30°C for 90 min. Samples were hard-spun in a pre-warmed, TLA-100 rotor (Beckman-Coulter), resuspend and analyzed by SDS-PAGE gel analysis.

## Yeast strains and plasmids

*Saccharomyces cerevisiae* strains used in this study are described in *Supplementary file 1* and are derivatives of SBY3 (W303). Construction of *stu2-3HA-IAA7* and a *LEU2* integrating plasmid containing wild-type *pSTU2-STU2-3V5* (pSB2232) are described in (*Miller et al., 2016*). *STU2* variants were constructed by mutagenizing pSB2232 as described in (*Liu and Naismith, 2008*; *Tseng et al., 2008*), resulting in pSB2254 (*pSTU2-STU2(Δ1–281)−3*V5, i.e. TOG1Δ), pSB2257 (*pSTU2-STU2(Δ282–550::GDGAGL)−3*V5, i.e. TOG2Δ), pSB2306 (*pSTU2-STU2(R200A)−3*V5), pSB2307 (*pSTU2-STU2(R519A)−3*V5), pSB2620 (*pSTU2-STU2(Δ12–245::321–559)−3*V5, i.e. TOG2-TOG2), pSB2817 (*pSTU2-STU2(Δ326–550::1–272)−3*V5, i.e. TOG1-TOG1).

## Spotting assay

For the spotting assay, the desired strains were grown overnight in yeast extract peptone plus 2% glucose (YPD) medium. The following day, cells were diluted to $OD_{600}$ ~1.0 from which a serial 1:5 dilution series was made and spotted on YPD + DMSO or YPD + 500 µM IAA (indole-3-acetic acid dissolved in DMSO) plates. Plates were incubated at 23°C for 3 days.

# Acknowledgements

We thank Jeff Woodruff for the generous gift of mCherry-Zyg-9, Susanne Bechstedt and Gary Brouhard for sharing a plasmid to express Stu2-GFP, and Jeff Moore for the gift of primers to construct an endogenous, tail-less tubulin yeast strain. LMR is the Thomas O Hicks Scholar in Medical Research. This work was supported by grants from the NIH (R01GM098543 to LMR and R01GM064386 to SB.) and the Robert A Welch Foundation (I-1908 to LMR). EAG was supported by NIH T32 GM008297, and by an NSF Graduate Research Fellowship, Grant No. 2014177758. This material is based upon work supported by the National Science Foundation Graduate Research Fellowship under Grant No. 2014177758. Any opinion, findings, and conclusions or recommendations expressed in this material are those of the authors(s) and do not necessarily reflect the views of the National Science Foundation. MPM is an HHMI Fellow of the Damon Runyon Cancer Research Foundation and SB is an investigator of the Howard Hughes Medical Institute.

# Additional information

## Funding

| Funder | Grant reference number | Author |
|---|---|---|
| National Institute of General Medical Sciences | R01 GM098543 | Luke M Rice |
| Welch Foundation | I-1908 | Luke M Rice |
| Howard Hughes Medical Institute | | Sue Biggins |
| National Institute of General Medical Sciences | R01 GM064386 | Sue Biggins |
| National Science Foundation | 2014177758 | Elisabeth A Geyer |
| National Institute of General Medical Sciences | T32 GM008297 | Elisabeth A Geyer |

The funders had no role in study design, data collection and interpretation, or the decision to submit the work for publication.

## Author contributions

Elisabeth A Geyer, Conceptualization, Formal analysis, Validation, Investigation, Methodology, Writing—original draft, Writing—review and editing; Matthew P Miller, Conceptualization, Investigation, Writing—review and editing; Chad A Brautigam, Investigation, Writing—review and editing; Sue Biggins, Conceptualization, Supervision, Funding acquisition, Writing—review and editing; Luke M Rice, Conceptualization, Data curation, Supervision, Funding acquisition, Investigation, Methodology, Writing—original draft, Project administration, Writing—review and editing

## Author ORCIDs

Matthew P Miller (iD) https://orcid.org/0000-0003-2012-7546
Sue Biggins (iD) http://orcid.org/0000-0002-4499-6319
Luke M Rice (iD) http://orcid.org/0000-0001-6551-3307

## Decision letter and Author response

Decision letter https://doi.org/10.7554/eLife.34574.026
Author response https://doi.org/10.7554/eLife.34574.027

# Additional files

## Supplementary files

• Supplementary file 1. Strains used in this study. All strains are derivatives of SBY3 (W303).
DOI: https://doi.org/10.7554/eLife.34574.021

• Transparent reporting form
DOI: https://doi.org/10.7554/eLife.34574.022

## Data availability

All data generated or analysed during this study are included in the manuscript

The following previously published dataset was used:

| Author(s) | Year | Dataset title | Dataset URL | Database, license, and accessibility information |
|---|---|---|---|---|
| Ayaz P, Ye X, Huddleston P, Brautigam CA, Rice LM | 2012 | A TOG:alpha/beta-tubulin Complex Structure Reveals Conformation-Based Mechanisms For a Microtubule Polymerase | http://www.rcsb.org/pdb/search/structid-Search.do?structureId=4FFB | Publicly available at the RCSB Protein Data Bank (accession no. 4FFB) |

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
