## [Decision Letter]

Thank you for submitting your article "Design principles of a microtubule polymerase" for consideration by *eLife*. Your article has been favorably evaluated by Andrea Musacchio (Senior Editor) and three reviewers, one of whom is a member of our Board of Reviewing Editors. The reviewers have opted to remain anonymous. As you will see below, the three reviewers are keen on publishing your paper, as long as you can make the suggested changes.

The reviewers have discussed the reviews with one another and the Reviewing Editor has drafted this decision to help you prepare a revised submission.

Summary:

The Stu2/XMAP215/chTOG family of proteins are key regulators of microtubule dynamics. They are potent catalysts of microtubule growth and achieve this through a combination of conserved, N-terminal, tubulin-binding TOG domains and additional domain(s) that can bind to the microtubule lattice. In "Design principles of a microtubule polymerase," Geyer et al. carefully examine how the different domains of *S. cerevisiae* Stu2 contribute to polymerase activity. The start off by showing that activity is basically proportional to number of active TOGs, in agreement with shown for Widlund et al. for XMAP. They make the unexpected observation that TOG identity plays a much smaller role compared to TOG position with respect to the basic microtubule lattice binding region(s). Furthermore, in a series of detailed experiments combining TOG and tubulin point mutants, they find that TOG-tubulin interaction appears to modulate lattice binding. Finally, using a tubulin mutant and epothilone B to stabilize or "straighten" the microtubule lattice, they show that Stu2 lattice binding is affected by tubulin curvature. Based on this data, they present a model where strong TOG:tubuin interaction is accompanied by weak lattice binding before tubulin incorporation. After release of tubulin into the growing microtubule end, lattice binding affinity increases. This results in the observed plus end specific localization of Stu2 which appears to require an active polymerase. As a whole, this work presents compelling data on the mechanism of action of an important family of microtubule regulators. The experiments are well designed and generally support the conclusions drawn.

Essential revisions:

You use point mutations in the TOG domains that argue that the number and type of TOGs is not important. However, to make the general claim that TOG is not particularly important, TOG2 would also need to be replaced with TOG1 in an identical experiment. Is there a reason this was not done? You do use a spot test in Figure 2—figure supplement 1C to show some activity of a TOG1-TOG1 construct in vivo. However, very similar rescue is seen with a TOG1-dTOG2 construct which was shown to be inactive in vitro (Figure 2D, E). Therefore, since the ability of these constructs to rescue in vivo do not appear to directly reflect polymerase activity in vitro, polymerase activity of TOG1-TOG1 construct (or any construct with TOG1 adjacent to the lattice binding domain) will need to be demonstrated. Otherwise, the claim as stated in the Abstract, "polymerase activity does not require different kinds of TOGs, nor are there strict requirements for how the TOGs are linked" will need to be modified.

We would also like you to clarify the affinity of TOG domains for mutant tubulins. What are the affinities of TOG domains for the tubulins with mutations (e.g. longitudinal contacts or lateral contacts)? Is it possible that differences in affinity cause the observed effects on Stu2-microtubule binding (Figure 4B)?

The way it is cited, the authors appear to suggest (subsection “Unpolymerized tubulin controls lattice binding by the basic region” –) that Brouhard et al., 2008 present evidence in support of a "tubulin shuttling" model. However, you present the contrary: XMAP215 does not act as a shuttle but as a tip tracking processive polymerase. In their model, XMAP215 (and bound tubulin) flux to the MT end was not fast enough to contribute to polymerization activity. If this was not the intention, please clarify.

Other points to consider:

A further experiment that was suggested was to investigate the distance between TOG domains and the basic region. It is unclear if the length/sequence of the linker connecting the TOG domain to the basic region is important for the activity. As this is a key point additional experiments are needed. Could the linker length be varied? Aligning sequences across different organisms may be helpful in the design of such constructs. Including these alignments in a supplementary table may also useful to readers. However although this would strengthen the paper, we see this as potentially beyond the scope of this paper, and leave this to your discretion.

The Reviewing Editor was also concerned about the extent to which Stu2 is processive and whether their experiments to measure the end-residence time of Stu2 are the right ones.

With 5 nM Stu2, you measured an end-residence time of ~2 seconds for a microtubule that is growing at ~0.5 μm/min. That growth rate converts to ~13 dimers per second per end, or 1 dimer per protofilament per second. This means that 2 second residence time isn't much for a protein with 4 TOG domains in the dimer. Although you point out that XMAP215 stayed on the end for ~4 seconds, the growth rate in those XMAP measurements was 3 μm/min or 82 dimers/sec/end. In other words, one could argue that Stu2 is not processive. Within those 2 seconds, it deposits its 4 tubulins and leaves.

But you are measuring the end residence time at 5 nM Stu2, a concentration which is basically the same as the "uncatalyzed rate". It might look different if you measured the residence time at 100 or 200 nM Stu2, when the microtubule is growing 5 times faster. Either the end-residence time will be much shorter (~0.4 seconds) or the processivity will be much higher. Those experiments are challenging because you need a small amount of tagged Stu2 alongside a higher concentration of untagged Stu2, or maybe you need to do FRAP and measure recovery rates.

This point is important for how you interpret the high affinity of Stu2 for microtubules after tubulin release. You argue the high affinity is a ratcheting mechanism that explains processivity. That can't be the main interpretation if there is no processivity.

The amount of Stu2 at the microtubule end seems to saturate before the growth rate effect saturates. More specifically, the Stu2 signal at the end hits max at 20 nM while the growth rate saturates at 60 nM. The authors describe these concentrations as equivalent, and I get that, but our working concentration range is 0200 nM effectively, so in fact it's a large difference. we wonder if this indicates that the tip structure of the microtubule end determines how effectively Stu2 can do its work. This might explain why the processivity numbers are so low – maybe Stu2 is not effective on a blunter microtubule end but works most effectively on an end that is tapered?

Please also discuss a model for how the proposed 'antagonism' between the basic region and the TOG domains could work.

---

## [Author Response]

Essential revisions:You use point mutations in the TOG domains that argue that the number and type of TOGs is not important. However, to make the general claim that TOG is not particularly important, TOG2 would also need to be replaced with TOG1 in an identical experiment. Is there a reason this was not done? You do use a spot test in Figure 2—figure supplement 1C to show some activity of a TOG1-TOG1 construct in vivo. However, very similar rescue is seen with a TOG1-dTOG2 construct which was shown to be inactive in vitro (Figure 2D, E). Therefore, since the ability of these constructs to rescue in vivo do not appear to directly reflect polymerase activity in vitro, polymerase activity of TOG1-TOG1 construct (or any construct with TOG1 adjacent to the lattice binding domain) will need to be demonstrated. Otherwise, the claim as stated in the Abstract, "polymerase activity does not require different kinds of TOGs, nor are there strict requirements for how the TOGs are linked" will need to be modified.

The reason we had not measured the activity of a Stu2 variant with a TOG1 domain in the basic-proximal position is that ‘transplanting’ a TOG1 domain turned out to be more sensitive to the domain boundaries than we had expected (or observed for TOG2). We have now succeeded in purifying a ‘TOG1-TOG1’ variant of Stu2, and we show that it has polymerase and end-tracking activity (new panel Figure 1E). The ‘trick’ to achieve this was to include some of the linker sequence (~250-272) that follows the structured core of the TOG1 domain (roughly amino acids 10-250). The TOG1-TOG1 polymerase was less active than a TOG2-TOG2 polymerase, perhaps reflecting the additional sequence between the end of the transplanted TOG1 domain and the beginning of the basic region (see the response to the first ‘other points to consider’ for more about this). In any event, these new data clearly demonstrate that a Stu2 variant that only contains TOG1 domains can exhibit polymerase activity. Thanks for the push to get these data.

We would also like you to clarify the affinity of TOG domains for mutant tubulins. What are the affinities of TOG domains for the tubulins with mutations (e.g. longitudinal contacts or lateral contacts)? Is it possible that differences in affinity cause the observed effects on Stu2-microtubule binding (Figure 4B)?

Excellent question. We now include quantitative binding experiments demonstrating that the TOG1 domain from Stu2 binds with comparable affinity to wild-type, ‘top-blocked’, and ‘side-blocked’ αβ-tubulin (new panels in Figure 5—figure supplement 1). More definitively than the gel filtration data we showed initially, these new experiments rule out mutation-induced changes in TOG-binding-affinity as a possible explanation for the effects we report in Figure 5. Chad Brautigam helped us collect and analyze these data, and consequently he has been added as an author to the revised paper.

The way it is cited, the authors appear to suggest (subsection “Unpolymerized tubulin controls lattice binding by the basic region”) that Brouhard et al., 2008 present evidence in support of a "tubulin shuttling" model. However, you present the contrary: XMAP215 does not act as a shuttle but as a tip tracking processive polymerase. In their model, XMAP215 (and bound tubulin) flux to the MT end was not fast enough to contribute to polymerization activity. If this was not the intention, please clarify.

Yes, that was a poorly placed citation. We deleted the first sentence of the paragraph (which contained the citation in question) and think the paragraph reads better now.

Other points to consider:A further experiment that was suggested was to investigate the distance between TOG domains and the basic region. It is unclear if the length/sequence of the linker connecting the TOG domain to the basic region is important for the activity. As this is a key point additional experiments are needed. Could the linker length be varied? Aligning sequences across different organisms may be helpful in the design of such constructs. Including these alignments in a supplementary table may also useful to readers. However although this would strengthen the paper, we see this as potentially beyond the scope of this paper, and leave this to your discretion.

We include one new experiment to begin addressing this question (New panels D, E in Figure 2—figure supplement 1). We purified a Stu2 variant containing a 16 amino acid flexible spacer (GSSGGSSSGSSGGGSG) in between the end of TOG2 (residue 560) and the start of the basic region (residue 561). This ‘spacer variant’ Stu2 displayed polymerase activity, but was demonstrably weaker than wild-type (or even TOG1-TOG2*) in terms of end-binding affinity and maximal fold stimulation of elongation. Along with the TOG1-TOG1 data described above, these new spacer data provide additional support for our claim that proximity between the basic region and a TOG domain is required.

The Reviewing Editor was also concerned about the extent to which Stu2 is processive and whether their experiments to measure the end-residence time of Stu2 are the right ones.

Thanks for the chance to clarify this. In retrospect we agree that we stumbled through our presentation of this important point.

With 5 nM Stu2, you measured an end-residence time of ~2 seconds for a microtubule that is growing at ~0.5 μm/min. That growth rate converts to ~13 dimers per second per end, or 1 dimer per protofilament per second. This means that 2 second residence time isn't much for a protein with 4 TOG domains in the dimer. Although you point out that XMAP215 stayed on the end for ~4 seconds, the growth rate in those XMAP measurements was 3 μm/min or 82 dimers/sec/end. In other words, one could argue that Stu2 is not processive. Within those 2 seconds, it deposits its 4 tubulins and leaves.But you are measuring the end residence time at 5 nM Stu2, a concentration which is basically the same as the "uncatalyzed rate". It might look different if you measured the residence time at 100 or 200 nM Stu2, when the microtubule is growing 5 times faster. Either the end-residence time will be much shorter (~0.4 seconds) or the processivity will be much higher. Those experiments are challenging because you need a small amount of tagged Stu2 alongside a higher concentration of untagged Stu2, or maybe you need to do FRAP and measure recovery rates.

We include new data about processivity where the Stu2 concentration is much higher (200 nM), and consequently much of the observed elongation can be attributed to the presence of Stu2. We used a mixture of 5 nM Stu2-GFP and 195 nM untagged Stu2 – this ‘spiking’ allowed us to retain ~single molecule sensitivity with much higher overall levels of Stu2. The distribution of end-residence times we measured for 200 nM ‘spiked’ Stu2 are very similar to what we measured for 5 nM Stu2-GFP. Thus, the end residence times do not decrease as elongation rate increases. This supports the claim that Stu2 is a processive polymerase (see below for a calculation).

This point is important for how you interpret the high affinity of Stu2 for microtubules after tubulin release. You argue the high affinity is a ratcheting mechanism that explains processivity. That can't be the main interpretation if there is no processivity.

Here is a simple calculation analogous to the one presented in the Brouhard 2008 paper about XMAP215. At 200 nM Stu2 the growth rate is 2.1 μm/min, which is 1.8 μm/min faster than the uncatalyzed growth rate of 0.3 μm/min. 1.8 μm/min of elongation attributable to Stu2 equates to ~49 dimers/second, or 98 dimers added during the ~2 second residence time for Stu2. We estimate there are about 6 Stu2 molecules on the end at saturation, so assuming they are all active we get 98/6 = 16 dimers added per Stu2. This is a similar order of magnitude to the ~25 dimers added per XMAP215 reported in Brouhard et al., 2008.

These new data allow us to discuss questions about end residence and processivity in more detail than we had in the initial manuscript.

The amount of Stu2 at the microtubule end seems to saturate before the growth rate effect saturates. More specifically, the Stu2 signal at the end hits max at 20 nM while the growth rate saturates at 60 nM. The authors describe these concentrations as equivalent, and I get that, but our working concentration range is 0200 nM effectively, so in fact it's a large difference. we wonder if this indicates that the tip structure of the microtubule end determines how effectively Stu2 can do its work. This might explain why the processivity numbers are so low – maybe Stu2 is not effective on a blunter microtubule end but works most effectively on an end that is tapered?

As we explained above, processivity only appeared to be low because we presented that section poorly. With the inclusion of new data and a better Discussion, we think the motivation for this question likely no longer applies.

Please also discuss a model for how the proposed 'antagonism' between the basic region and the TOG domains could work.

The mechanistic origin of this antagonism remain mysterious to us. We added some speculation about how it might be working in the Discussion.